# Tespa1 regulates T cell receptor-induced calcium signals by recruiting inositol 1,4,5-trisphosphate receptors

Jingjing Liang[1,2,*], Jun Lyu[1,2,*], Meng Zhao[3], Dan Li[1,2], Mingzhu Zheng[1,2], Yan Fang[4], Fangzhu Zhao[1], Jun Lou[1,2], Chuansheng Guo[1,2], Lie Wang[1], Di Wang[1,2], Wanli Liu[3] & Linrong Lu[1,2,5,6,7]

Thymocyte-expressed, positive selection-associated 1 (Tespa1) is important in T cell receptor (TCR)-driven thymocyte development. Downstream of the TCR, Tespa1 is a crucial component of the linker for activation of T cells (LAT) signalosome, facilitating calcium signalling and subsequent MAPK activation. However, it is unknown how Tespa1 elicits calcium signalling. Here, we show that inositol 1,4,5-trisphosphate receptor type 1 (IP3R1) is crucial for Tespa1-optimized, TCR-induced $Ca^{2+}$ flux and thymocyte development. Upon TCR stimulation, Tespa1 directly interacts with IP3R1 and recruits it to the TCR complex, where IP3R1 is phosphorylated at Y353 by Fyn. This Tespa1-IP3R1 interaction is mediated by the F187 and F188 residues of Tespa1 and the amino-terminus of IP3R1. Tespa1-F187A/F188A mutant mice phenocopy Tespa1-deficient mice with impaired late thymocyte development due to reduced IP3R1 translocation to the TCR-proximal region. Our work elucidates the function of Tespa1 in T cell development and the regulation of TCR-induced $Ca^{2+}$ signalling through IP3R1.

[1] Institute of Immunology, Zhejiang University School of Medicine, Hangzhou 310058, China. [2] Program in Molecular and Cellular Biology, Zhejiang University School of Medicine, Hangzhou 310058, China. [3] Ministry of Education Key Laboratory of Protein Sciences, Collaborative Innovation Center for Diagnosis and Treatment of Infectious Diseases, Institute for Immunology, School of Life Sciences, Tsinghua University, Beijing 100084, China. [4] Ministry of Education Key Laboratory of Protein Sciences, Collaborative Innovation Center for Diagnosis and Treatment of Infectious Diseases, Institute for Immunology, School of Life Sciences, Tsinghua University, Beijing 100084, China. [5] ZJU-UoE Institute, Zhejiang University School of Medicine, Hangzhou 310058, China. [6] Innovation Center for Cell Signaling Network, Zhejiang University School of Medicine, Hangzhou 310058, China. [7] Dr Li Dak Sum & Yip Yio Chin Center for Stem Cell and Regenerative Medicine, Zhejiang University School of Medicine, Hangzhou 310058, China. * These authors contributed equally to this work. Correspondence and requests for materials should be addressed to L.L. (email: lu_linrong@zju.edu.cn) or to W.L. (email: liulab@tsinghua.edu.cn).

Stimulation of the T cell receptor (TCR) triggers activation of the Src family protein tyrosine kinases Lck and Fyn, leading to the recruitment and activation of zeta chain-associated protein kinase 70 (ZAP70). Activated ZAP70 cooperates with Lck to phosphorylate the adaptor protein linker of activated T cells (LAT), which in turn recruits multiple signalling proteins, including phospholipase C gamma 1 (PLC- γ1)[1]. The subsequent recruitment of interleukin-2-induced tyrosine kinase (Itk) triggers the tyrosine phosphorylation and activation of PLC-γ1, which hydrolyses phosphatidylinositol-4,5-bisphosphate (PIP2) to produce the second messengers diacylglycerol (DAG) and inositol 1,4,5-trisphosphate (IP3). DAG predominantly activates the nuclear factor-κB signalling pathway via activation of protein kinase C θ and the Ras-mediated signalling pathway[2]. On the other hand, IP3 binds and activates IP3 receptors (IP3Rs), $Ca^{2+}$-permeable ion channels on the endoplasmic reticulum (ER) membrane, and triggers $Ca^{2+}$ release from the ER. The decreased $Ca^{2+}$ concentration in the ER evokes the activation of $Ca^{2+}$-release activated channels on the plasma membrane, leading to the sustained $Ca^{2+}$ influx necessary for subsequent activation of the transcription factor NFAT (nuclear factor of activated T cells) and the expression of related cytokines[3,4].

Although $Ca^{2+}$ flux is a signalling event that occurs secondary to PLC-γ1 activation, it is one of the fastest responses to TCR activation, occurring within 1 min in the TCR-proximal region[5]. This speed can be explained by the earlier finding that IP3R1 and TCR co-localize within the macromolecular LAT signalling complex upon LAT phosphorylation and PLC-γ1 activation[6,7]. Moreover, clustering of IP$_3$R1 at the TCR-proximal region induces the Y353 phosphorylation of IP3R1 by Fyn, which leads to a fivefold increase in affinity for IP3, in addition to reduced $Ca^{2+}$-dependent inactivation of the IP3R1 channel[8]. The phosphorylation of IP3R1 at Y353 is thus a critical signalling event for optimal $Ca^{2+}$ release and subsequent NFAT activation, which are crucial for T cell activation[7]. However, the mechanism by which IP3R1 is recruited to the TCR-proximal region is not clear, and the physiological relevance of this interaction in T cells is unknown.

Thymocyte-expressed, positive selection-associated 1 (Tespa1) was originally identified as a critical signalling molecule in thymocyte development[9]. Tespa1 deficiency impairs thymocyte positive selection, as reflected by fewer mature thymic and peripheral CD4$^+$ and CD8$^+$ T cells. Tespa1 associates with the LAT signalosome upon TCR activation and participates in the TCR-driven activation of the ERK-AP-1 and $Ca^{2+}$-NFAT pathways. The similarity of Tespa1 to Ki-Ras-induced actin-interacting protein (KRAP) in a conserved PFF motif led to the prediction that Tespa1 would interact with IP3R (ref. 10), and it has been reported that human Tespa1 protein interacts with IP3R1 and regulates $Ca^{2+}$ signalling[11]. To further understand the function of Tespa1 in TCR signalling, we perform a mass spectrometric analysis of proteins interacting with Tespa1 in Jurkat cells. In addition to many known TCR signalling molecules, we detect all members of the IP3R family of proteins, suggesting a potential role of the Tespa1-IP3R1 interaction in mediating the TCR-induced $Ca^{2+}$ signalling cascade. In this study, our results demonstrate that Tespa1 can directly bind to PLC-γ1 and IP3R1, thereby facilitating TCR-induced calcium signalling and thymocyte development.

## Results

### Tespa1 interacts with the X/Y catalytic domain of PLC-γ1.
To determine the direct binding partners of Tespa1 in the LAT signalosome, we cloned the major signalling molecules and evaluated whether they bound to Tespa1 in a yeast two-hybrid system; and we found that Tespa1 directly interacted with the X catalytic domain of PLC-γ1 (Supplementary Fig. 1). We next constructed a series of PLC-γ1 deletion mutants to further clarify the structural motifs of PLC-γ1 responsible for this interaction in HEK293 cells (Fig. 1a). Although co-immunoprecipitation revealed only a weak interaction between Tespa1 and full-length PLC-γ1, this interaction was significantly enhanced after deletion of either the X or the Y catalytic domain (PLC-γ1-ΔX or PLC-γ1-ΔY) (Fig. 1b). It is well known that PLC-γ1 undergoes a conformational change upon activation: in the resting state, the cSH2 domain interacts with the TIM-barrel catalytic domain and covers the substrate binding site, upon activation, phosphorylation of tyrosine 783 releases the cSH2 domain and uncovers the TIM-barrel catalytic domain[12]. We suspected that deletion of either of the catalytic domains may release its counterpart for binding to Tespa1. We thus tested this idea using the cSH2 deletion mutant. Indeed, deletion of cSH2 enhanced the interaction between Tespa1 and PLC-γ1 (Fig. 1b). We further confirmed this interaction with PLC-γ1 harbouring the point mutations E347K (in the X catalytic domain), Y747/R748E (in the cSH2 domain) and D1019K (in the Y catalytic domain), all of which disrupt the interaction between the TIM-barrel domain and the cSH2 domain[13], and found that all these mutations enhanced the interaction (Fig. 1c). This finding is in agreement with our observation that Tespa1 binds PLC-γ1 upon TCR activation in Jurkat T cells (Fig. 1d). In summary, these results suggest that the recruitment of Tespa1 to the LAT signalosome upon TCR activation might be mediated by the interaction between Tespa1 and PLC-γ1.

### Tespa1 directly interacts with IP3R1.
To further clarify the mechanism by which Tespa1 regulates TCR signalling, we established a Jurkat cell line that stably expressed Flag-tagged Tespa1. Flag-tagged Tespa1 and its interacting proteins were pulled down from whole-cell lysates of anti-CD3 plus anti-CD28-stimulated Jurkat cells using anti-Flag beads. The precipitates were then subjected to mass spectrometry. In addition to many known components of TCR signalling cascades, Tespa1 interacted with IP3R1, 2 and 3, a family of ER proteins (Supplementary Fig. 2a). Interestingly, this interaction, and the importance of the residues F187 and F188 in mediating the interaction, was predicted based on the sequence similarity of Tespa1 to KRAP[10]. We next verified the interaction between Tespa1 and IP3Rs and the role of the corresponding F187 and F188 sites in mediating this interaction. In HEK293 cells, overexpressed Tespa1 interacted with IP3R1-N (amino acids 1–610) in an F187- and F188-dependent manner (Fig. 2a,b). Similarly, wild-type Tespa1, but not the F187A/F188A mutant, interacted with endogenous IP3R1 in Jurkat cells (Fig. 2c). We further verified the direct interaction between Tespa1 and IP3R1-N in a two-hybrid system (Supplementary Fig. 2b,c).

### Tespa1 recruits IP3R1 to TCR-proximal region.
IP3R1, despite its primary location on the ER membrane, has been reported to redistribute upon TCR activation and co-localize with TCR complexes[7]. We thus speculated that Tespa1 might serve as an adaptor protein to recruit IP3R1 to TCR complexes. To test this hypothesis, we established Jurkat cell lines overexpressing either wild-type Tespa1 (J-Tespa1) or the Tespa1-F187A/F188A mutant (J-FFAA) and monitored the translocation of IP3R1 in these cells after TCR stimulation. Confocal microscopy showed a dispersed pattern of IP3R1 inside the cell in both cell lines in the resting state. However, after TCR stimulation for 5 min, IP3R1 staining was redistributed and localized in proximity to LAT in J-Tespa1 cells but not in J-FFAA cells (Fig. 3a).

To further confirm the differential localization of IP3R1 in the presence of Tespa1 or Tespa1-FFAA without the influence of endogenous Tespa1, we generated transgenic mice expressing the Flag-tagged Tespa1-FFAA mutant under control of the human CD2 promoter[14], bred the mutant mice onto a $Tespa1^{-/-}$ background, and then analysed the expression of Flag-tagged

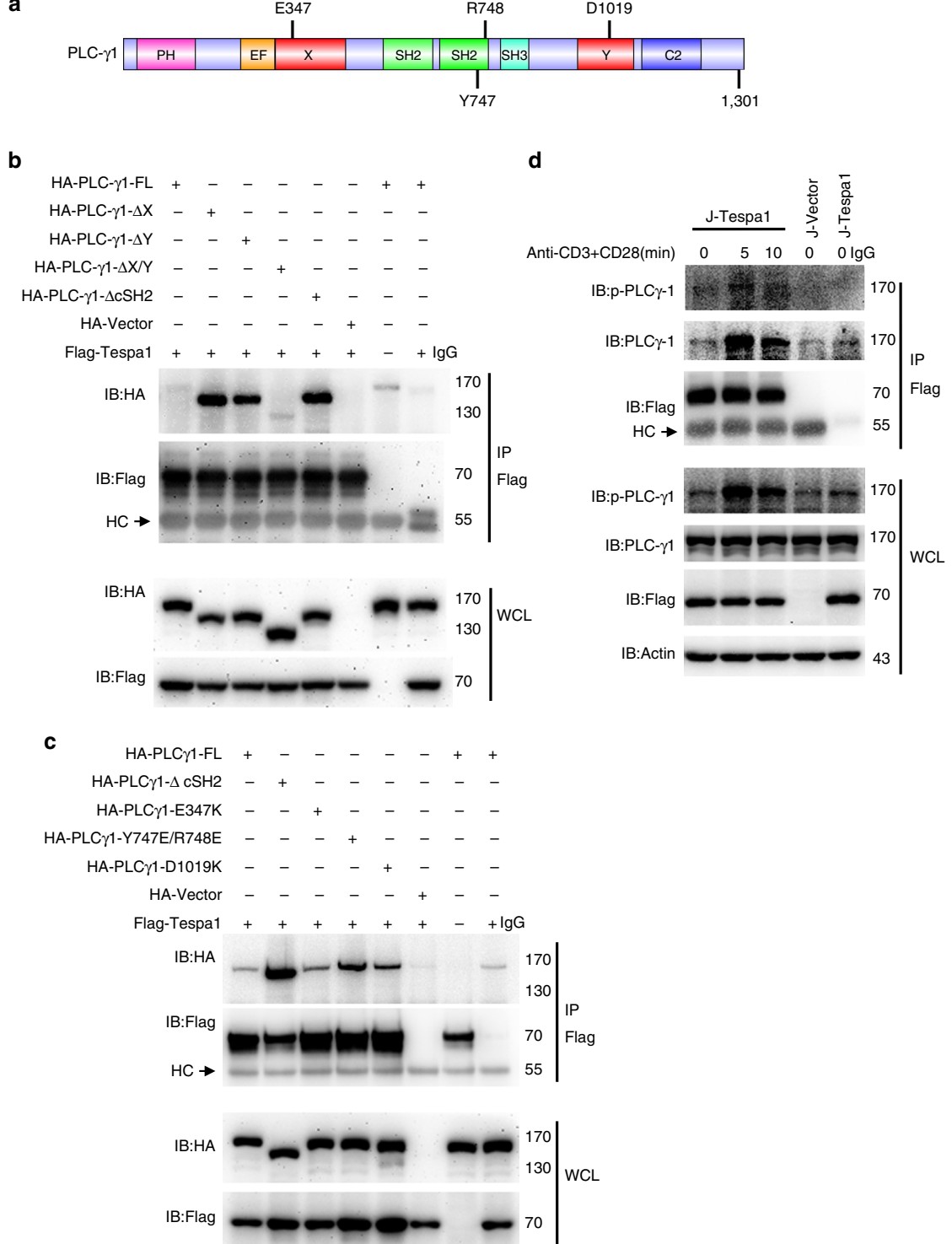

**Figure 1 | Association of PLC-γ1 and Tespa1.** (**a**) Schematic of the functional domains of PLC-γ1. (**b**) Immunoblot analysis (IB) of HEK293 cells co-transfected with Flag-tagged Tespa1 and HA-tagged full-length PLC-γ1 (FL) or its mutants. Cells were lysed, immunoprecipitated (IP) with anti-Flag beads, and probed with anti-Flag and anti-HA. Bottom, immunoblot analysis of whole-cell lysates (WCL, without immunoprecipitation). HC, heavy chain. (**c**) HA-tagged full-length PLC-γ1 (FL) or PLC-γ1 with a point mutation in the X catalytic domain, cSH2 domain or Y catalytic domain was co-expressed and analysed for interaction with Tespa1. (**d**) Immunoblot analysis of Jurkat cells transfected with Flag-tagged Tespa1 (J-Tespa1) or Flag empty vector (J-Vector). Cells were left unstimulated (0) or were stimulated with anti-CD3 and anti-CD28 antibodies for 5 or 10 min, following by lysis, immunoprecipitation with anti-Flag beads and probing with anti-PLC-γ1 or anti-p-PLC-γ1 antibodies. Data are representative of at least three experiments.

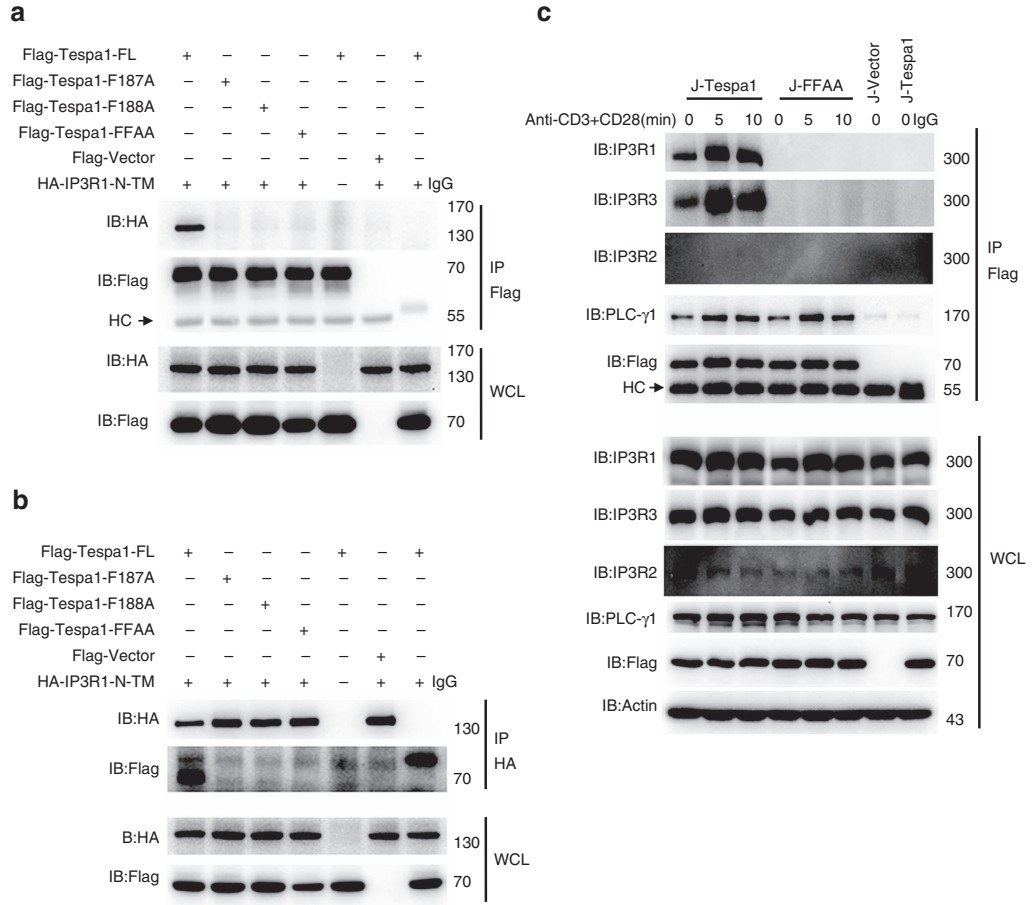

**Figure 2 | Interaction between Tespa1 and IP3R1.** (**a,b**) Immunoblot analysis (IB) of HEK293 cells co-transfected with Flag-tagged full-length Tespa1 (FL) or different Tespa1 mutants and HA-tagged IP3R1-N-TM. Cells were lysed and immunoprecipitated (IP) with anti-Flag beads (**a**) or anti-HA antibodies plus protein G beads (**b**), then probed with anti-Flag and anti-HA antibodies. Bottom, immunoblot analysis of whole-cell lysates (WCL, without immunoprecipitation). (**c**) Immunoblot analysis of Jurkat cells transfected with Flag-tagged Tespa1 (J-Tespa1), Flag-tagged Tespa1-F187A/F188A (J-FFAA) or Flag empty vector (J-Vector). Cells were left unstimulated (0) or were stimulated with anti-CD3 and anti-CD28 antibodies for 5 or 10 min, lysed, immunoprecipitated with anti-Flag beads and probed with anti-IP3R1, anti-IP3R2 or anti-IP3R3 antibodies. Data are representative of at least three experiments.

Tespa1-FFAA by immunoblotting (Supplementary Fig. 3). We used structured illumination microscopy (SIM) to capture the relocalization of IP3R1 around the plasma membrane at the interface between the cells and anti-CD3 coated coverslips. Although the 3D display (Fig. 3b) showed that there were no significant changes in the overall IP3R1 distribution in the whole cells after TCR activation, Tespa1-dependent recruitment of IP3R1 appeared to occur at the interface after stimulation (Fig. 3c,d). We also performed total internal reflection fluorescence microscopy (TIRFM) experiments and observed the recruitment of IP3R1 into the interface between the WT T cell and the anti-CD3 coated coverslips, which were severely impaired in activated $Tespa1^{-/-}$ and $Tespa1$-FFAA T cells (Supplementary Fig. 4). Interestingly, the recruitment of IP3R into the interface is not totally absent in both $Tespa1^{-/-}$ and $Tespa1$-FFAA cells. The mildly enhanced staining signals in these cells upon anti-CD3 antibody treatment might be caused by closer contact of the cell to the coverslips upon antibody binding. All these data together suggested the potential role of Tespa1 in mediating the relocalization of IP3R1 to TCR-proximal region after TCR activation.

**Tespa1 promotes the Y353 phosphorylation of IP3R1.** The Y353 phosphorylation of IP3R1 is also critical for $Ca^{2+}$ signalling in Jurkat cells[7]. Therefore, we assessed the Y353 phosphorylation of

IP3R1 in J-Tespa1 and J-FFAA cells using confocal microscopy. Unstimulated J-Tespa1 and J-FFAA cells showed dispersed, weak signals indicating basal Y353 phosphorylation. Upon TCR activation, the immunofluorescence signal of IP3R1 phospho-Y353 increased significantly and relocalized closer to LAT in J-Tespa1 cells but not in J-FFAA cells (Fig. 4a). SIM also confirmed that wild-type thymocytes showed more distinct relocalization of IP3R1 phospho-Y353 closer to TCR complexes than Tespa1-deficient and $Tespa1$-FFAA thymocytes (Fig. 4b,c). In addition, the TIRFM experiment also revealed defective IP3R1 phospho-Y353 accumulation at the interface between T cells and anti-CD3 coated coverslips in $Tespa1^{-/-}$ and $Tespa1$-FFAA thymocytes (Supplementary Fig. 5).

To investigate whether Y353 phosphorylation of IP3R1 is dependent on Tespa1-mediated IP3R1 translocation, we next constructed two IP3R1 mutants with different subcellular localization patterns and co-expressed them with the active form of Fyn kinase (Y528F) in HEK293 cells. The IP3R1-N (amino acids 1–610) mutant, which resides freely in the cytoplasm, was highly phosphorylated at Y353 by Fyn, independent of its binding with Tespa1. However, IP3R1-N-TM, which contains the transmembrane domain (TM) and localizes to the ER, exhibited much less Y353 phosphorylation. More importantly, Y353 phosphorylation of IP3R1-N-TM was severely impaired when Tespa1 was absent or could not bind to

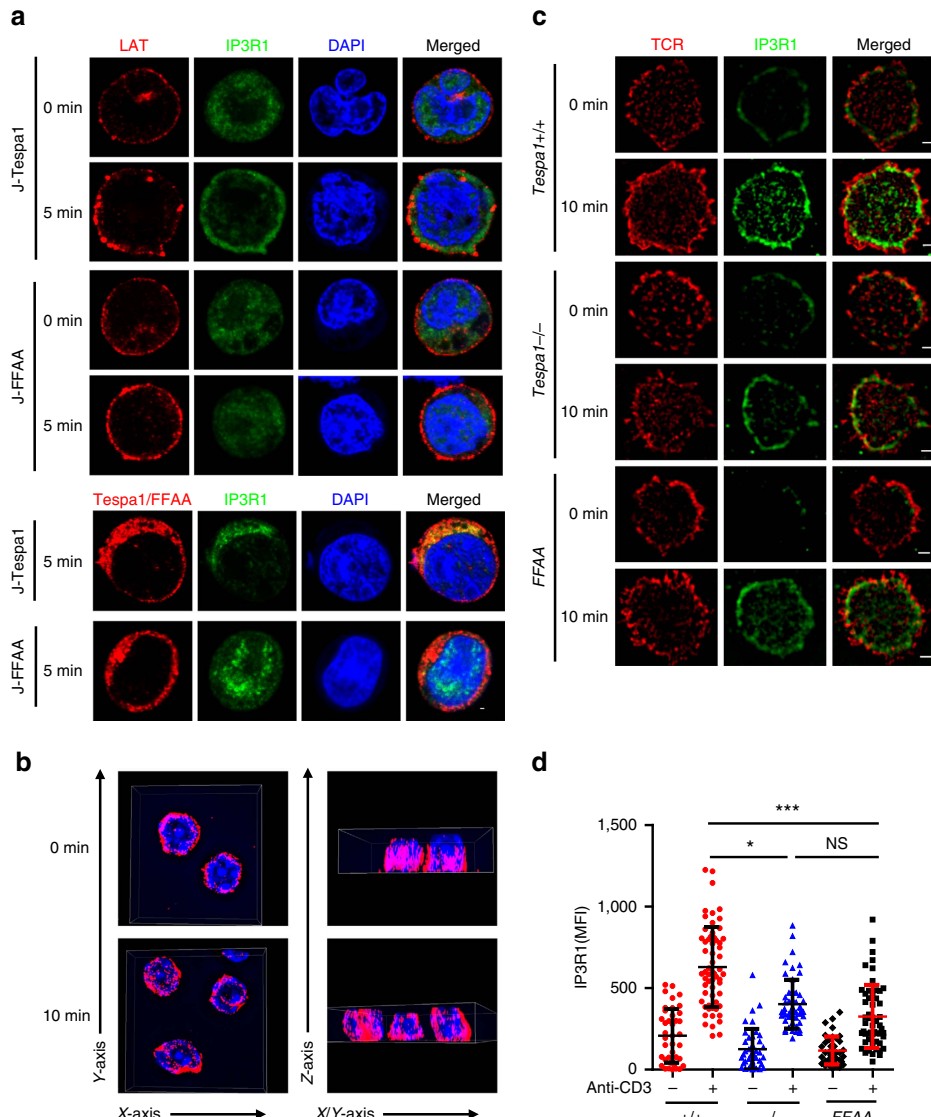

**Figure 3 | Recruitment of IP3R1 to TCR-proximal region by Tespa1.** (**a**) Confocal microscopy images of J-Tespa1 and J-FFAA cells labelled with anti-LAT (red), anti-IP3R1 (green), and DAPI (blue) before (0 min) and after (5 min) stimulation with soluble anti-CD3 and anti-CD28 antibodies. Scale bar, 1 μm. (**b**) WT thymocytes before (0 min) and after (10 min) stimulation with anti-CD3 immobilized on coverslips were stained with anti-IP3R1 antibodies, and then a 4 μm Z-stack 3D display was taken from contacting coverslip by Structured illumination microscopy (SIM). Left, a view from side of contacting coverslip. Right, a view upright to Z-axis. (**c**) SIM images of thymocytes from $Tespa1^{+/+}$ ($+/+$), $Tespa1^{-/-}$ ($-/-$) and $Tespa1$-FFAA (FFAA) mice labelled with anti-IP3R1 (green) and Alexa Fluor 647-conjugated anti-TCR β (red) before (0 min) and after (10 min) stimulation with immobilized anti-CD3 antibodies (10 μg ml$^{-1}$). Scale bar, 1 μm. (**d**) Mean fluorescence intensity (MFI) of IP3R1 from SIM analysis as in **b**. Each symbol represents an individual cell; small horizontal lines indicate the mean ± s.d. One-way ANOVA was performed with a P value included. *$P < 0.05$, ***$P < 0.001$, NS, not significant. Data are representative of at least three experiments.

IP3R1-N-TM (Fig. 4d). These results suggested that free IP3R1-N can easily interact with plasma membrane-localized Fyn for Y353 phosphorylation, whereas it is difficult for ER-located IP3R1-N-TM to access Fyn and be phosphorylated, except after being recruited to the nearby membrane region via binding to Tespa1. All these results suggested that Tespa1 is critical for the redistribution of IP3R1 and the Fyn-mediated Y353 phosphorylation of IP3R1.

**Tespa1-IP3R1 interaction facilitates calcium signalling.** Having previously shown that Tespa1 is essential for optimal Ca$^{2+}$ signalling in response to TCR activation[9], we speculated that the binding of Tespa1 with IP3R1 might be responsible for this

function. We first assayed TCR-induced Ca$^{2+}$ flux in J-Tespa1 and J-FFAA cells and found that the former showed a much higher TCR-induced Ca$^{2+}$ flux than the latter (Fig. 5a). Consistently, exogenous wild-type Tesap1, but not Tespa1-FFAA, led to higher TCR-induced NFAT activity and more extensive NFAT1 nuclear localization (Fig. 5b and Supplementary Fig. 6). All these results indicated that the interaction between Tespa1 and IP3R1 is a critical step in the activation of the TCR-induced Ca$^{2+}$-NFAT signalling pathway.

$Tespa1$-FFAA double-positive (DP) and CD4$^{+}$CD8$^{int}$ thymocytes, exhibited impaired TCR- and mitogen-driven Ca$^{2+}$ release from the ER, comparable to that in $Tespa1^{-/-}$ cells (Fig. 5c,d), while store-operated Ca$^{2+}$ influx did not differ among $Tespa1^{+/+}$, $Tespa1^{-/-}$ and $Tespa1$-FFAA thymocytes.

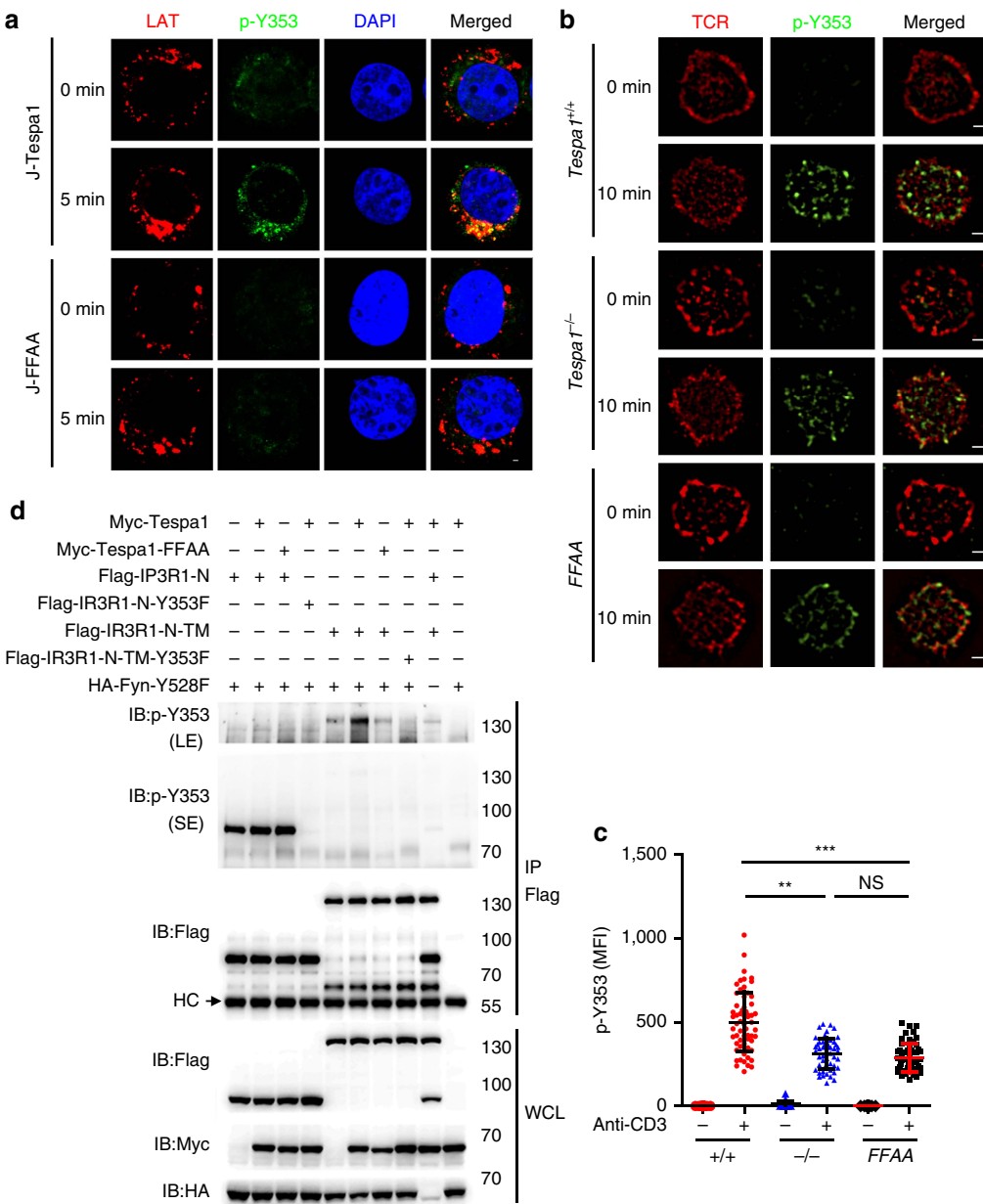

**Figure 4 | Requirement of Tespa1 for Y353 phosphorylation of IP3R1.** (**a**) Confocal microscopy images of J-Tespa1 or J-FFAA cells labelled with anti-LAT (red), anti-IP3R1 phospho-Y353 (p-Y353, green) and DAPI (blue) before (0 min) and after (5 min) stimulation with soluble anti-CD3 and anti-CD28 antibodies. Scale bar, 1 μm. (**b**) Structured illumination microscopy (SIM) images of thymocytes from *Tespa1*$^{+/+}$ (+/+), *Tespa1*$^{-/-}$ (−/−) and *Tespa1-FFAA* (*FFAA*) mice labelled with anti-IP3R1 phospho-Y353 (p-Y353, green) and Alexa Fluor 647-conjugated anti-TCR α/β (red) before (0 min) and after (10 min) stimulation with immobilized anti-CD3 antibodies (10 μg ml$^{-1}$). Scale bar, 1 μm. (**c**) Mean fluorescence intensity (MFI) of IP3R1 phospho-Y353, assessed by SIM analysis as in **b**. Each symbol represents an individual cell; small horizontal lines indicate the mean ± s.d. One-way ANOVA was performed with a *P* value included. \*\**P* < 0.01, \*\*\**P* < 0.001, NS, not significant. (**d**) HEK293 cells were transiently transfected with combinations of plasmids expressing Flag-tagged IP3R1-N, IP3R1-N-Y353F, IP3R1-N-TM or IP3R1-N-TM-Y353F, HA-tagged Fyn-Y528F, Myc-tagged Tespa1 or Tespa1-FFAA. Flag-tagged proteins were pulled down with anti-Flag beads and probed with anti-IP3R1 phospho-Y353 (p-Y353) antibody by immunoblotting. p-Y353 (SE) denotes a short exposure of the membrane probed for phospho-Y353 of IP3R1-N and IP3R1-N-Y353F. p-Y353 (LE) denotes a long exposure of membrane probed for phospho-Y353 of IP3R1-N-TM and IP3R1-N-TM-Y353F. The IP3R1-N-TM/IP3R1-N-TM-Y353F phospho-Y353 band was removed from the membrane containing IP3R1-N and IP3R1-N-Y353F and exposed separately for long time. IP3R1-N-TM is a fusion protein consisting of IP3R1-N (amino acids 1–610) and the transmembrane domain (TM) (amino acids 2216–2749) of IP3R1 that is located on the ER. Data are representative of at least three experiments.

Similarly, the lower TCR- or mitogen-mediated luciferase reporter activity for the transcription factors AP-1 and NFAT in sorted *Tespa1-FFAA* DP thymocytes remained the same as that in *Tespa1*$^{-/-}$ cells (Fig. 5e). Since it has been reported that altered IP3R function might result in disrupted cellular Ca$^{2+}$ homeostasis and decreased intracellular Ca$^{2+}$ stores[15,16],

reduced basal IP3R1 activation in Tespa1-deficient cells might be responsible for this phenomenon. However, we could not rule out the possibility that Tespa1 might have other unidentified functions in regulating the ER calcium store or be involved in PMA-ionomycin-induced calcium flux through an unknown mechanism.

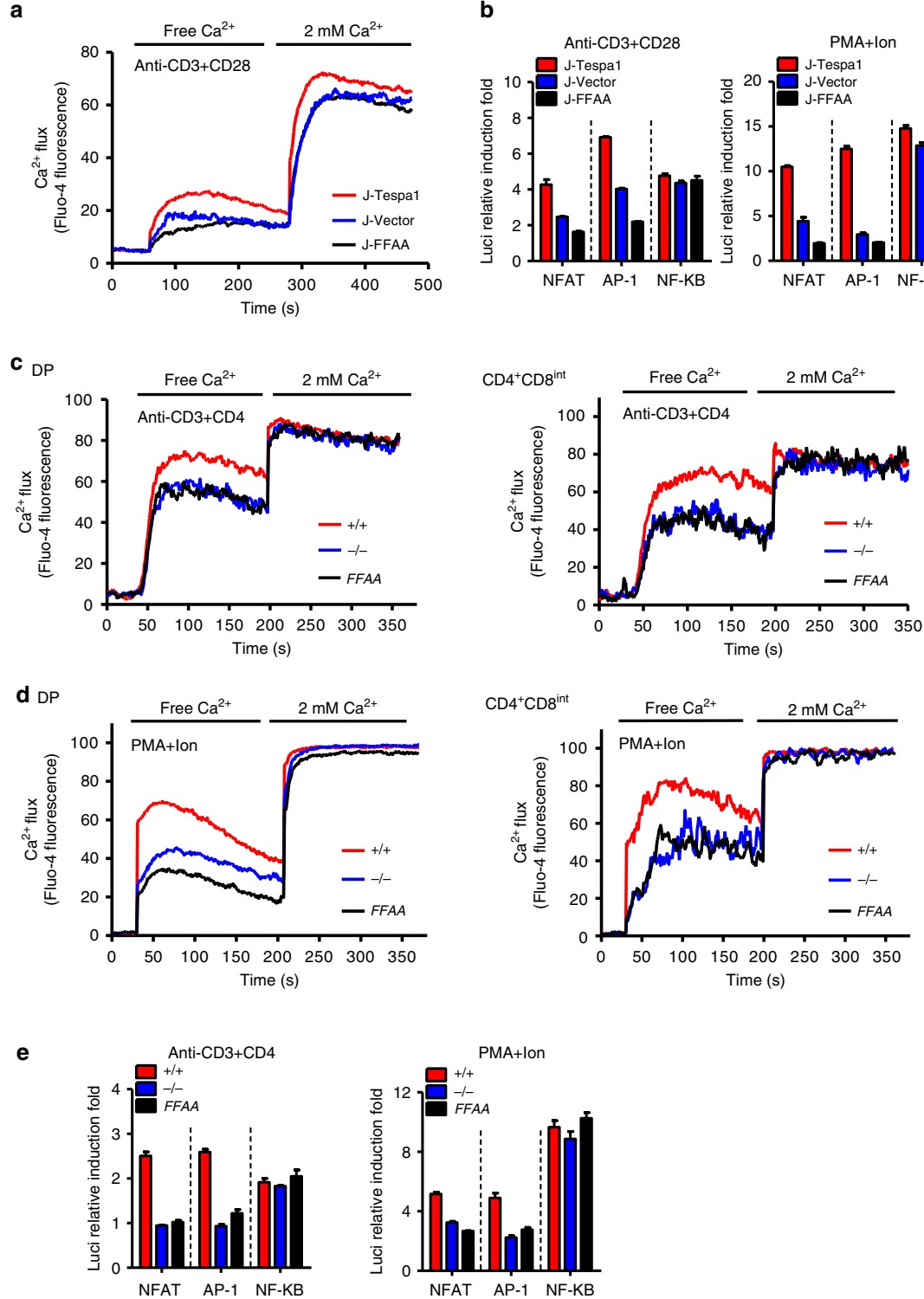

**Figure 5 | Tespa1-IP3R1 interaction facilitates calcium signalling.** (**a**) Recordings of $Ca^{2+}$ flux in J-Tespa1, J-FFAA, and J-Vector cells. Cells were stimulated by crosslinking biotinylated anti-CD3 and anti-CD28 antibodies (anti-CD3+CD28) with streptavidin in $Ca^{2+}$-free medium, then, 2 mM $CaCl_2$ was added. (**b**) Luciferase activity in J-Tespa1, J-FFAA and J-Vector cells transfected with luciferase reporter constructs (horizontal line). Cells were allowed to 'rest' for 30 min, and were then stimulated with anti-CD3 plus anti-CD28 antibodies (left) or PMA plus ionomycin (right) for 6 h, results were normalized to renilla luciferase activity and are presented relative to those of unstimulated cells. (**c,d**) Recordings of $Ca^{2+}$ flux from $Tespa1^{+/+}$ (+/+), $Tespa1^{-/-}$ (−/−) and $Tespa1$-FFAA (FFAA) thymocytes. Cells were stimulated by crosslinking biotinylated anti-CD3 and anti-CD4 antibodies (anti-CD3+CD4, **c**) with streptavidin or by PMA and ionomycin (PMA+Ion, **d**) in $Ca^{2+}$-free medium, then, 2 mM $CaCl_2$ was added. (**e**) Luciferase activity in sorted DP thymocytes from $Tespa1^{+/+}$ (+/+), $Tespa1^{-/-}$ (−/−) and $Tespa1$-FFAA (FFAA) mice. Cells were transfected with luciferase reporter constructs (horizontal line), allowed to 'rest' for 30 min, and then stimulated with anti-CD3 and anti-CD4 antibodies (left) or PMA plus ionomycin (right) for 6 h, results were normalized to renilla luciferase activity and are presented relative to those of unstimulated cells. Data are representative of at least three experiments (error bars indicate s.d. in **b**,**e**).

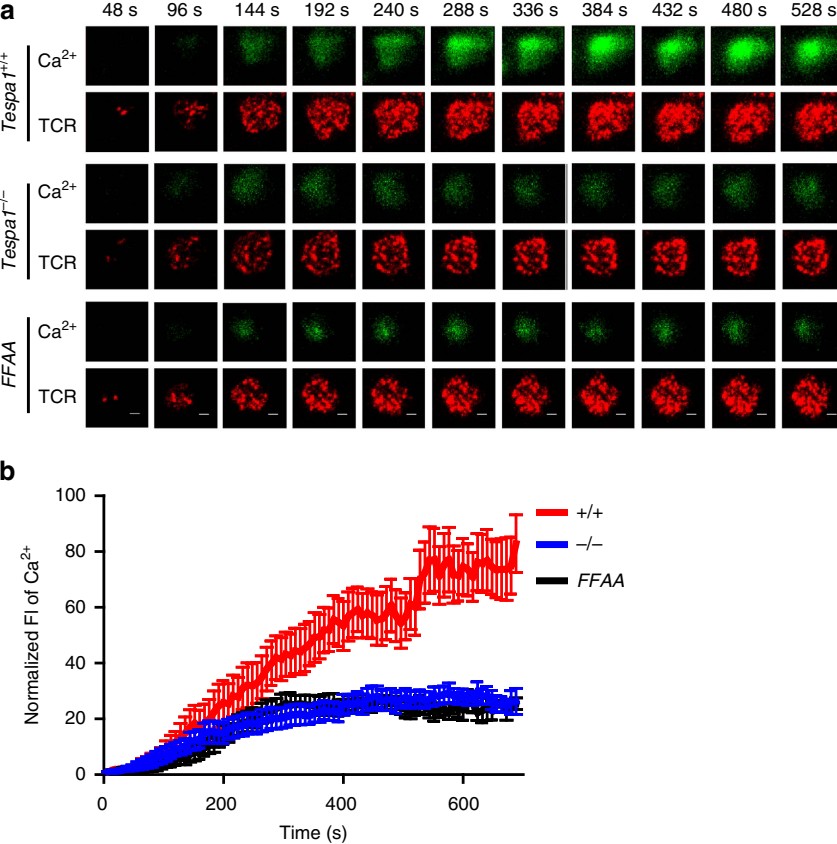

**Figure 6 | Impaired TCR-proximal calcium flux in *Tespa1-FFAA* T cells.** (**a**) Two-colour time-lapse TIRFM images of Fluo4 (green) and Alexa Fluor 647-Fab anti-TCR α/β (red) captured at the indicated time points (Supplementary Movie 1) from isolated *Tespa1*$^{+/+}$, *Tespa1*$^{-/-}$ and *Tespa1-FFAA* (*FFAA*) DP cells placed on planar lipid bilayers containing anti-CD3 antibodies. See Supplementary Movie 1 for the complete time-lapse TIRFM images. Scale bar, 3 μm. (**b**) Normalized fluorescence intensity (FI) of Ca$^{2+}$ imaging (Fluo4) from high-speed, high-resolution, time-lapse TIRFM images of real-time Ca$^{2+}$ flux in the TCR-proximal region in isolated *Tespa1*$^{+/+}$ (+/+), *Tespa1*$^{-/-}$ (−/−) and *Tespa1-FFAA* (*FFAA*) DP cells as in **a**. Data are representative of at least three experiments. Bars represent mean ± s.e.m. of 10 (*FFAA*) or 12 (+/+ and −/−) cells from one representative of three independent experiments.

We also analysed and compared the activation of other signalling pathways in *Tespa1*$^{+/+}$, *Tespa1*$^{-/-}$ and *Tespa1-FFAA* DP thymocytes activated by anti-CD3 plus anti-CD4 antibodies. Consistent with our previous results, we found no difference in the phosphorylation of PLC-γ1, p38 or JNK. Interestingly, the decreased phosphorylation of ERK in *Tespa1*$^{-/-}$ thymocytes was not restored by Tespa1-FFAA reconstitution (Supplementary Fig. 7), indicating that IP3R1-Tespa1 binding is indispensable for ERK activation. This impairment in ERK activation can be explained by the defective Ca$^{2+}$ flux in *Tespa1-FFAA* DP thymocytes[17].

**Relocation of IP3R1 ensures rapid TCR-proximal calcium flux.** TCR-induced Ca$^{2+}$ flux is a rapid signalling event that initiates within 1 min in the TCR-proximal region[5]. The recruitment of IP3R1 to the TCR-proximal region by Tespa1 is likely part of this event, as it favours a localized and faster Ca$^{2+}$ flux. We next used time-lapse TIRFM to capture the Ca$^{2+}$ signal specifically in the TCR-proximal region[18]. All the Ca$^{2+}$ imaging experiments were performed in Ca$^{2+}$-free medium to exclude the influence of store-operated calcium channels (SOCs) or other Ca$^{2+}$ channels on the plasma membrane; thus, the Ca$^{2+}$ signal captured in this experiment was mediated only by IP3Rs. The loss of Tespa1-IP3R1 binding in both *Tespa1*$^{-/-}$ and *Tespa1-FFAA* DP cells led to a much lower Ca$^{2+}$ signal around the TCR complexes (Fig. 6a,b and Supplementary Movie 1). This result indicated the

importance of the Tespa1-IP3R1 interaction in initiating a rapid, localized Ca$^{2+}$ signal in the TCR-proximal region.

**Tespa1 regulates thymocyte development through IP3R1.** The newly generated *Tespa1-FFAA* mice enabled us to assess directly whether the Tespa1-IP3R1 axis is the major mechanism by which Tespa1 regulates thymocyte development. Surprisingly, the phenotype of *Tespa1-FFAA* mice was identical to that of Tespa1-knockout mice. The *Tespa1-FFAA* mice showed reductions in both CD4 and CD8 single-positive (SP) thymocytes similar to those in *Tespa1*$^{-/-}$ mice (Fig. 7a). In addition, perturbation of Tespa1-IP3R1 binding severely impaired the maturation of CD4 and CD8SP thymocytes (Fig. 7b). Because upregulation of CD69 expression is a critical marker of successful positive selection[19,20], we next evaluated the surface expression of CD69 on *Tespa1*$^{+/+}$, *Tespa1*$^{-/-}$ and *Tespa1-FFAA* thymocytes. Reconstitution of the Tespa1-FFAA protein also failed to rescue the downregulated surface expression of CD69 in Tespa1-deficient DP and CD4$^+$CD8$^{int}$ cells (Fig. 7c). To determine the specific stage at which thymocyte development was blocked in *Tespa1-FFAA* mice, we quantified cells at five distinct developmental stages defined by the expression of TCRβ and the activation marker CD69. We found comparable numbers of immature TCR$^{lo}$CD69$^{lo}$ cells (population 1; mostly double-negative and DP cells) and TCR$^{int}$CD69$^{lo}$ cells (population 2; preselection DP cells) in *Tespa1-FFAA*, *Tespa1*$^{-/-}$ mice and wild-type mice. However,

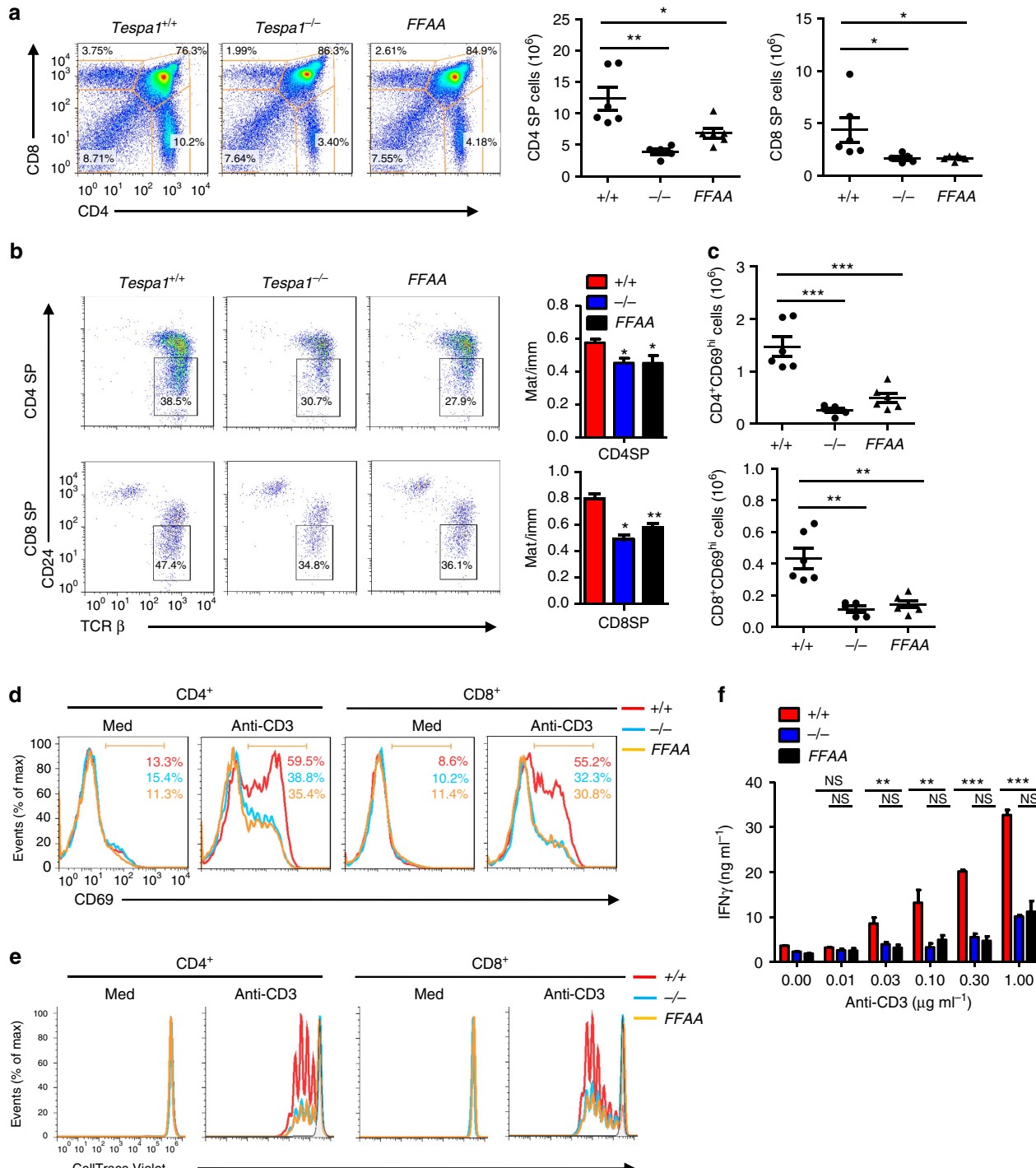

**Figure 7 | Impaired positive selection in *Tespa1-FFAA* mice.** (**a**) Surface staining of CD4 and CD8 (left) on *Tespa1*$^{+/+}$ (+/+), *Tespa1*$^{-/-}$ (−/−) and *Tespa1-FFAA* (*FFAA*) thymocytes. The numbers in or adjacent to the outlined areas (or in quadrants) indicate the percentage of cells in each area throughout. Right, quantification of CD4$^+$ (CD4SP) and CD8$^+$ (CD8SP) thymocyte subpopulations. (**b**) Surface expression of TCR-β and CD24 on the gated CD4SP or CD8SP thymocytes. The numbers adjacent to the outlined areas indicate the percentage of cells in the CD24$^{lo}$TCRβ$^{hi}$ gate. Right, ratio of mature (CD24$^{neg-lo}$) cells to immature (CD24$^{hi}$) cells (Mat/imm). (**c**) Quantification of CD4$^+$CD69$^{hi}$ and CD8$^+$CD69$^{hi}$ cells from the surface staining of CD4, CD8 and CD69 on *Tespa1*$^{+/+}$ (+/+), *Tespa1*$^{-/-}$ (−/−) and *Tespa1-FFAA* (*FFAA*) thymocytes. (**d**) CD69 staining on the gated CD4$^+$ or CD8$^+$ cells from Tespa1$^{+/+}$ (+/+), Tespa1$^{-/-}$ (−/−) and *Tespa1-FFAA* (*FFAA*) total splenocytes left unstimulated (Med) or stimulated by TCR crosslinking with anti-CD3 antibodies for 6 h. The numbers below the bracketed lines indicate the percentage of CD69$^+$ cells (the colours match the key). (**e**) Flow cytometric analysis of proliferation by CellTrace Violet staining in sorted *Tespa1*$^{+/+}$ (+/+), *Tespa1*$^{-/-}$ (−/−) and *Tespa1-FFAA* (*FFAA*) CD4$^+$ and CD8$^+$ splenocytes left unstimulated (Med) or stimulated with anti-CD3 antibodies for 72 h *in vitro*. (**f**) Statistics for enzyme-linked immunosorbent assays detecting the level of secreted IFN-γ by sorted *Tespa1*$^{+/+}$ (+/+), *Tespa1*$^{-/-}$ (−/−) and *Tespa1-FFAA* (*FFAA*) CD4$^+$ splenocytes after a 3 days *in vitro* stimulation with different doses of plate-bound anti-CD3 antibodies. Each symbol (**a**,**c**) represents an individual mouse; small horizontal lines indicate the mean (and s.d.). *$P < 0.05$, **$P < 0.01$, ***$P < 0.001$, and NS, not significant (two-tailed *t*-test). Data are representative of three independent experiments with five or six mice per group (**a**,**c**) (mean ± s.d. in **a**,**c**, error bars indicate s.d. in **b**,**f**).

Tespa1-FFAA and Tespa1$^{-/-}$ mice had significantly fewer cells in population 3 (thymocytes undergoing selection), population 4 (post-positive-selection thymocytes) and population 5 (mature SP cells ready for export to the periphery) (Supplementary Fig. 8).

Accordingly, the Tespa1-FFAA mice had fewer peripheral T cells in the spleen, similar to the effect in Tespa1-deficient mice (Supplementary Fig. 9a). Peripheral T cells from both Tespa1-FFAA and Tespa1$^{-/-}$ mice had greater frequencies of memory (CD44$^{hi}$CD62L$^{lo}$) T cells than wild-type mice (Supplementary Fig. 9b). Consistently, splenic CD4$^+$ and CD8$^+$ T cells in Tespa1-FFAA and Tespa1$^{-/-}$ mice exhibited more BrdU uptake in vivo, reflecting the homeostatic expansion of lymphocytes in these partially lymphopenic mice (Supplementary Fig. 9c)[21].

Consistent with our previous findings in Tespa1-deficient mice, the defect in thymocyte development in Tespa1-FFAA mice was accompanied by diminished TCR-mediated responses. We found defective TCR-driven CD69 upregulation in Tespa1-FFAA CD4$^+$ T cells and CD8$^+$ T cells, comparable to that in Tespa1$^{-/-}$ T cells (Fig. 7d). Tespa1-FFAA reconstitution failed to rescue the impaired proliferative capacity and cytokine production of Tespa1$^{-/-}$ CD4$^+$ and CD8$^+$ T cells in response to anti-CD3 stimulation (Fig. 7e,f). These data indicated that the thymocyte development phenotype of Tespa1$^{-/-}$ mice is largely due to disruption of the Tespa1-IP3R1 interaction.

## Discussion

Although the critical role of Tespa1 in regulating TCR signalling and thymocyte development was demonstrated in our previous study, the molecular mechanisms remained unknown[9]. The identification of F187 and F188 from sequence analysis and the ability of human Tespa1 to bind to IP3Rs suggested a direct role of Tespa1 in regulating IP3Rs (refs 10,11). Indeed, Tespa1 must bind to IP3R1 through its F187 and F188 residues to mediate optimal Ca$^{2+}$ flux upon TCR activation, and the loss of IP3R1 binding is responsible for the decreased Ca$^{2+}$ signal and ERK activation in Tespa1$^{-/-}$ thymocytes. More importantly, the Tespa1-FFAA mutant failed to rescue the thymocyte development defect in Tespa1$^{-/-}$ mice. Thus, our results have uncovered the molecular mechanisms by which Tespa1 regulates thymocyte development.

Our study also revealed the role of Tespa1 as an adaptor protein between PLC-γ1 and IP3R1, enabling the recruitment of IP3R1 to the TCR-proximal region. Previous studies have reported that IP3R1 re-localizes with the TCR complex upon T cell activation[6]. The translocation of IP3R1 influences ligand-induced Ca$^{2+}$ flux in both T and B cells[22]. Subsequent studies have shown that IP3R1 co-clusters with the TCR complex immediately after TCR stimulation and is phosphorylated at Y353 by Fyn[7,8]. However, the precise mechanism by which IP3R1 redistributes and co-localizes with the TCR complex remains elusive. Our results suggest that the direct interaction of Tespa1 with IP3R1 mediates the recruitment of IP3R1 to the TCR complex after TCR activation.

It has been reported that Y353 phosphorylation increases the affinity of IP3R1 for IP3 and that this is responsible for full activation of the Ca$^{2+}$ signal[7]. We also showed that the recruitment of IP3R1 to the TCR complex is essential for the Y353 phosphorylation of IP3R1, as disruption of the Tespa1-IP3R1 interaction diminished this phosphorylation. This is consistent with the important role of the Tespa1-IP3R1 interaction in mediating Ca$^{2+}$ activation found in our study. In addition, the Tespa1-mediated recruitment of IP3R1 to TCR complexes brings IP3R1 into close proximity to a local intracellular microenvironment with elevated IP3 levels generated by activated PLC-γ1 upon TCR stimulation, thus

ensuring faster and more efficient activation of IP3R1, which in turn boosts the Ca$^{2+}$ signal at the TCR-proximal region. Although the recruitment and phosphorylation of IP3R1 upon TCR activation have been previously reported, these studies were mostly carried out in cultured cells. The physiological relevance of IP3R1 translocation remains unclear. Thus, our data from the Tespa1-FFAA mice also demonstrated the importance of IP3R relocation in regulating thymocyte development.

It has also been reported that Tespa1 interacts with GRP75 and regulates TCR-induced Ca$^{2+}$ flux into mitochondria[23]. This is based on the finding that Tespa1 partially co-localizes with the ER and mitochondria in resting Jurkat cells. However, the knockdown or knockout of Tespa1 affects Ca$^{2+}$ flux into both the cytoplasm and mitochondria. Thus the reduced Ca$^{2+}$ flux into mitochondria may reflect a secondary effect due to decreased IP3Rs activity in general.

In addition to T cells, mouse mast cells also express high levels of Tespa1. We have previously shown that bone marrow-derived mast cells (BMMCs) from Tespa1-deficient mice have much higher levels of FcεRI (high-affinity immunoglobulin E receptor)-elicited higher Ca$^{2+}$ flux than wild-type BMMCs, contrary to the impaired TCR-induced Ca$^{2+}$ flux in DP cells from Tespa1-FFAA and Tespa1$^{-/-}$ mice. This inconsistency might be explained by the preferential association of Tespa1 with LAT2 in mast cells, which mediates the suppression of FcεRI signalling by recruiting phosphatases, including SHP-1 (ref. 24). This could be clarified by investigating the co-localization of IP3R1 and LAT2 and the phosphorylation status of IP3R1 in mast cells.

In summary, our results elucidated the mechanism by which Tespa1 regulates IP3R1 function in TCR-driven Ca$^{2+}$ signalling and the essential role of the Tespa1-IP3R axis in thymocyte development.

## Methods

**Mice.** Tespa1$^{-/-}$ mice were generated by homologous recombination- mediated gene targeting at the Shanghai Research Center for Model Organisms as previously described[25]. Mice on a mixed 129 × C57BL/6 background were back-crossed onto the C57BL/6 background for 6–8 generations. To generate Tespa1 transgenic mice, PCR-cloned cDNA fragments encoding the F187A/F188A mutant of Tespa1 were inserted into the hCD2-VA vector[14]. The transgenes were purified and microinjected into the pronuclei of fertilized eggs from C57BL/6 mice using standard procedures. The embryos were transferred to the oviducts of pseudopregnant ICR mice. Previous studies have shown that the human CD2 promoter/enhancer directs the expression of transgenes in the T cell lineage in mice. Each transgenic line was then bred with Tespa1$^{-/-}$ mice to replace endogenous wild-type Tespa1 with the Tespa1-FFAA mutant. Age-matched littermate mice used for experimental purpose were between 4 and 6 weeks of age. All experimental animal protocols were approved by the Review Committee of Zhejiang University School of Medicine and were in compliance with institutional guidelines.

**Antibodies.** The biotin anti-mouse CD3 (145-2C11), biotin anti-mouse CD4 (GK1.5), anti-mouse CD28 (37.51), anti-human CD3 (OKT3) and anti-human CD28 (CD28.2) antibodies were from Biolegend (San Diego, CA, USA). The Phospho-PLCγ1 (Tyr783) Antibody (#2821, pAb,1:1,000), IP3 Receptor 1 Antibody (D53A5, mAb,1:1,000), p44/42 MAPK (Erk1/2) Antibody (#9102, pAb,1:1,000), Phospho-p44/42 MAPK (Erk1/2) (Thr202/Tyr204) Antibody (#9101, pAb,1:1,000), Phospho-SAPK/JNK (Thr183/Tyr185) Antibody (#9251, pAb,1:1,000), JNK2 Antibody (56G8, mAb,1:1,000), p38 MAPK Antibody (#9212, pAb,1:1,000), Phospho-p38 MAPK (Thr180/Tyr182) Antibody (#9211, pAb,1:1,000) were from Cell Signaling Technology (Danvers, MA, USA). Anti-Phospholipase C gamma 1 (EP1898-7Y, mAb,1:1,000), Anti-SLP76 (EPR2549, mAb,1:1,000), anti-phospholipase C gamma 1 phospho-Y783 (EPR2611Y, mAb,1:1,000) and anti-IP3R1 (#ab5804, pAb,1:1,000) antibodies were from Abcam (Cambridge, MA, USA). The anti-GADS (#06-983, pAb,1:1,000) and anti-Lck (#06-583, pAb,1:1,000) antibodies were from Merck Millipro (Billerica, MA, USA). The specific antibody against phospho-Y353 IP3R1 was used at 1:100 dilutions and was generated by HuaAn Biotechnology (Hangzhou, China). The polyclonal antibody against mouse Tespa1 was used at 1:500 dilutions and was generated by HuaAn Biotechnology using the GST-fused mouse Tespa1 recombinant protein fragment (amino acids 2–182). Alexa Fluor 647-conjugated goat anti-TCR-β chain (H57-597, Biolegend) was fragmented using a Fab micro preparation kit (Pierce, Rockford, IL), according to the manufacturer's protocol[26].

**Plasmid constructs and transfections.** Mouse Tespa1 was amplified by PCR and inserted into the pMSCV puro plasmid (Clontech, Kusatsu, Shiga, Japan), at the XhoI and EcoRI restriction enzyme sites. The pMSCV puro plasmid containing the mouse Tespa1-F187A/F188A mutant was generated using a Site-directed Gene Mutagenesis Kit (Beyotime, Suzhou, China). To generate stable cell lines, Jurkat cells were infected with packaged retrovirus harbouring pMSCV puro-Tespa1, pMSCV puro-Tespa1-FFAA or pMSCV puro and selected for growth in $4 \mu g \, ml^{-1}$ puromycin. pEGFP-N1 plasmids containing the IP3R1-N terminus (amino acids 1–610) were kindly provided by Professor Senji Shirasawa. Using these plasmids as templates, the IP3R1- N terminus was amplified by PCR and inserted into pIPHA2 or pIPFlag2, at the EcoRI and XhoI restriction enzyme sites. The IP3R1-N-Y353F mutant was generated as described above.

**Immunoprecipitation and immunoblot analysis.** Cells were lysed in NP-40 lysis buffer containing 50 mM Tris (pH 7.4), 150 mM NaCl, 1% NP-40, 10 mM phenylmethylsulfonyl fluoride (PMSF), 20 mM NaF, 1 mM $Na_3VO_4$ and protease inhibitor 'cocktail' (Sigma-Aldrich, St Louis, MO, USA). The lysates were immunoprecipitated with the appropriate antibodies for 4 h at 4 °C. Protein A/G Sepharose beads (Roche, Basel, Switzerland) were then added and the samples were incubated overnight. Alternatively, lysates were immunoprecipitated with anti-Flag M2 magnetic beads (Sigma-Aldrich) for 4 h at 4 °C. After three washes, the samples were resolved using SDS-PAGE gels (10%) and blotted onto membranes. For immunoblot analysis, cells were lysed in SDS sample buffer by the addition of 1/4 volume of $5 \times$ SDS sample buffer directly into cell suspensions. The samples were then boiled for 5 min and separated by 10% SDS-PAGE. The full-size images for all the immunoblot analysis are presented in Supplementary Figs 11-12 in the Supplementary Information.

**Intracellular immunofluorescence staining and molecular imaging.** Recruitment of IP3R1 and IP3R1 p-Y353 into the interface between T cells and anti-CD3 coated coverslips was imaged by tTIRFM following published protocol with modifications[26]. In brief, T cells were labelled with 200 nM Fab fragment of Alexa Flour 647-conjugated anti-mouse TCR beta Abs. The labelled cells were washed twice and then loaded to the coverslips containing anti-CD3 that cross-link TCR. Cells were fixed with 4% paraformaldehyde for 10 min after incubation with the anti-CD3 at 37 °C, 5% $CO_2$. The fixed T cells were permeabilized with 0.1% Triton X-100 and pretreated with blocking reagent containing $100 \, g \, ml^{-1}$ donkey nonspecific IgG (Jackson Immunoresearch Laboratory) and 1% BSA. Subsequently, cells were stained with anti-IP3R1 Ab (Abcam) or anti-IP3R1 p-Y353, followed by secondary Ab Alexa Fluor 555-conjugated donkey Abs specific for rabbit IgG (Invitrogen) staining as previously described. Images were analysed by ImageJ (NIH) software following published protocols[26]. In brief, images were subtracted for background and then marked with regions of interest (ROIs). The mFI values obtained were the ratio of integrated FI of the ROIs to the total area of pixels in the same ROIs. The display range of a set of TIRFM images in each figure is the same to allow direct visional comparison of intensity.

**Immunofluorescence and confocal microscopy.** Cells were fixed in prewarmed 4% paraformaldehyde for 30 min and permeabilized with 0.2% Triton X-100 for 10 min. After blocking with 5% BSA, cells were stained overnight with anti-LAT and anti-IP3R1 or anti-IP3R1 p-Y353. Staining was detected using DyLight 488- and DyLight 549-labelled secondary antibodies (Multiscience). Nuclei were co-stained with DAPI (Roche). Stained cells were viewed under a confocal fluorescence microscope (IX81-FV1000; Olympus).

**Structured illumination microscopy.** Coverslips were treated with a 0.01% w v$^{-1}$ poly-L-lysine solution (Sigma-Aldrich) for 5 min, drained, and dried at RT for 15 min. Purified anti-mouse CD3 antibodies (145-2C11, Biolegend) in PBS ($20 \mu g \, ml^{-1}$) was incubated with the coverslips at 4 °C overnight. Coverslips were rinsed once in PBS before use. Thymocytes ($1 \times 10^6$) in RPMI-1640 were plated on the coverslips, which were then incubated on ice for 15 min. The cells were stimulated for 10 min by incubating the coverslips at 37 °C. Then, the cells were fixed with 4% paraformaldehyde in PBS for 15 min at room temperature and stained with Alexa Fluor 647-conjugated anti-TCR antibodies at room temperature for 20 min. The cells were washed and permeabilized with 0.2% Triton X-100 in PBS for 10 min. Then, non-specific binding was blocked by incubation with PBS plus 2% BSA for 30 min at room temperature. Next, the samples were incubated with anti-IP3R1 or anti-IP3R1 p-Y353 antibodies at room temperature for 4 h. The coverslips were washed with PBS five times (5 min each time) and incubated with Cy3B-labelled goat anti-rabbit IgG antibody for 1 h at room temperature. After the coverslips were washed, nuclei were stained with DAPI, and the coverslips were mounted with a drop of mounting medium. Images were obtained with a 3D structural illumination microscope equipped with a $\times 100$ objective lens with laser excitation at 405, 488, 561 and 640 nm. For image acquisition, the focal plane was adjusted to the interface of the cells and the coverslips as described in Supplementary Fig. 10. For z-stack analysis, optical sections were obtained from the interface along the Z-axis at 0.2 μm intervals. All images and z-stacks were reconstructed by NIS Elements AR with the following parameters: illumination modulation contrast, 0.5; high-resolution noise suppression, 1; out-of-focus

blur suppression, 0.15. Images were analysed with MetaMorph. The mean fluorescence intensity (MFI) was calculated using the following equation:
MFI = (total intensity)/ROI.

**$Ca^{2+}$ flux.** Thymocytes in suspension were first labelled with $4 \mu g \, ml^{-1}$ Fluo4 (Invitrogen) for 1 h at 37 °C, washed with ice-cold PBS and re-suspended in PBS. Cells were surface labelled with phycoerythrin-indocarbocyanine-conjugated anti-CD4 and allophycocyanin-conjugated anti-CD8 for 30 min on ice and then incubated with biotinylated anti-CD3 ($5 \mu g \, ml^{-1}$) and anti-CD4 ($5 \mu g \, ml^{-1}$). The labelled cells were warmed for 20 min at room temperature and then crosslinked with streptavidin ($25 \mu g \, ml^{-1}$) or stimulated with PMA (phorbol 12-myristate 13-acetate) and ionomycin immediately before flow cytometry analysis. Mean fluorescence ratios were plotted after analysis with FlowJo software (TreeStar).

**$Ca^{2+}$ imaging.** Sorted DP cells were first loaded with $4 \mu g \, ml^{-1}$ Fluo4, suspended in Hanks' balanced salt solution and then surface stained with Alexa Fluor 647-Fab anti-TCR α/β antibody for 5 min on ice. The cells were placed on fluid planar lipid bilayers containing anti-CD3 immediately before imaging. TIRFM images were acquired every 8 s at 37 °C on the heated stage of an Olympus IX81 microscope supported by a TIRF port, an ANDOR iXon+ DU897D electron-multiplying charge-coupled device camera (ANDOR Technology) and an Olympus $\times 100$ 1.45 numerical aperture lens. A 488-nm/514-nm argon gas laser and a 568-nm/647-nm red krypton/argon gas laser were equipped and used as indicated. The acquisition was controlled by MetaMorph software (Molecular Devices) and the exposure time was 250 ms. The acquired images were analysed and processed with Image-Pro Plus (Media Cybernetics, Silver Spring, MD) and Image J (National Institutes of Health, USA) following published protocols[26].

**T cell activation and proliferation assay.** Splenocytes from Tespa1$^{+/+}$, Tespa1$^{-/-}$ or Tespa1-FFAA mice were cultured for 5 h in the presence of plate-bound anti-CD3 in round-bottom 96-well plates, stained with anti-CD69, anti-CD4 and anti-CD8 antibodies, and analysed by flow cytometry. For proliferation assays, cells were stained with 0.5 μM CellTrace Violet (Invitrogen, New York, NY, USA) for 30 min at 37 °C. Staining was stopped by the addition of an equal volume of RPMI. After washing, cells were cultured for 72 h with or without different doses of anti-CD3 in RPMI. T cell proliferation was measured by analysing CellTrace Violet dilution by flow cytometry.

**Statistical testing.** Group mean values were compared by two-tailed Student's $t$-test or one-way ANOVA. $P$ values of less than 0.05 were considered significant. Statistics were calculated using GraphPad Prism 5 software (GraphPad Software Inc.).

**Data availability.** The data that support the findings of this study are available from the corresponding author upon request.

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

## Acknowledgements

The authors thank Dr Chenqi Xu (Shanghai Institutes for Biological Sciences, Chinese Academy of Sciences) and Lie Wang (Zhejiang University) for helpful discussion and Chun Guo, Shuangshuang Liu, Guifeng Xiao, Hangjun Wu, Zhaoxiaonan Lin, Jiajia Wang and Yingying Huang from the core facilities of the Zhejiang University School of Management for technical assistance with microscopy and FACS analysis. This work was supported by grants from the National Natural Science Foundation of China (31325009, 31530019 and 31270927 to L.L. 31500708 to M.Z.), the National Basic Research Program of China (973 Program) (2012CB945004 and 2011CB944100 to L.L.), the Zhejiang Provincial Natural Science Foundation of China (LR14H100001 to D.W.), the Project Founded by China Postdoctoral Science Foundation (2015M580516, 2016T90546 to M.Z.), the Fundamental Research Funds for the Central Universities (2016QNA7020 to M.Z.), and the Program of Introducing Talents of Discipline to Universities (B13026 L.L.). L.L. also received funding support from the Specialized Research Fund for the Doctoral Program of Higher Education (20130101110004) and the Zhejiang University K.P. Chao High Technology Development Foundation.

## Author contributions

Ji.L., Ju.L., W.L., D.W. and L.L. designed the study; Ji.L., Ju.L., M.Z., D.L., Mi.Z., Y.F., Ju.L. F.Z. and C.G. performed the experiments; Ji.L., Ju.L., D.L. and L.L. analysed the data; Ji.L., Ju.L., W.L. and L.L. wrote the paper.

## Additional information

**Competing interests:** The authors declare no competing financial interests.

DOI: 10.1038/ncomms16183

# Author Correction: Tespa1 regulates T cell receptor-induced calcium signals by recruiting inositol 1,4,5-trisphosphate receptors

Jingjing Liang, Jun Lyu, Meng Zhao, Dan Li, Mingzhu Zheng, Yan Fang, Fangzhu Zhao, Jun Lou, Chuansheng Guo, Lie Wang, Di Wang, Wanli Liu & Linrong Lu

*Nature Communications* **8**:15732 doi: 10.1038/ncomms15732 (2017); Published online 9 Jun 2017; Updated 12 Mar 2018

In the original version of this Article, the affiliation details for Yan Fang were incorrectly given as "ZJU-UoE Institute, Zhejiang University School of Medicine, Hangzhou 310058, China". This has now been corrected in both the PDF and HTML versions of the Article.

1