## [Peer Review File · Nature Communications]

Reviewers' comments:

Reviewer #1 (Remarks to the Author):

This is a potentially interesting and novel study. The authors use a range of approaches to demonstrate the role of Tespa1 in regulating TCR-mediated calcium signals. However, it was not an easy read and not fully convincing; the figures were not well described in the text, and the figure legends did not describe the experimental data at all well. I had to stare at the figures for a while to work out what was going on and correlate it with the authors' statements. I am not sure that many people will give the manuscript so much time. For example, Panel d in figure 1 is entirely explained by the sentence (line 126) 'This finding supported our previous finding that Tespa1 binds to the LAT signalosome upon TCR activation (Fig. 1d).' The data is not solely a previous finding! It's in the current manuscript, and it needs to be described!

Some of the figures are not labelled correctly. For example, Panels a and b in Figure 2 have the same labelling, but the IPs and blots were performed with different antibodies (to show a co-IP). Please check the figures carefully. It's very confusing for the reader otherwise.

The authors may have discovered a novel role for Tespa1 in linking signalling proteins downstream of TCR stimulation, but I have really struggled with the manuscript in terms of data quality, explanation of how data were obtained, explanation of the figures and control experiments.

Major comments:

1. I don't find the immunoblots in Figure 1 to be fully convincing of the points the authors make. On line 128, the authors state 'In sum, these results demonstrated that the recruitment of Tespa1 to the LAT signalosome upon TCR activation is mediated by interaction with the exposed catalytic domain of PLC- γ 1.' However, this is not directly shown by the data. In fact, there is little (or none) interaction of full-length PLC- γ 1 with Tespa1 according to the data shown. Only deletion mutations, or point mutations, increase binding. Whilst this is consistent with the authors' suggestions, it is not proof. Does co-IP of over-expressed proteins in HEK cells does replicate TCR organisation in thymocytes? I'm not convinced it does. Panel d in figure 1 indicates a receptor-activated interaction that is again consistent with the authors' suggestion. However, I think the authors should tone down their conclusion from these data.
2. Was there are difference in the Tespa1 interactome in unstimulated versus stimulated cells? The main text does not say so, but the legend to Supplementary Figure 2 states that cells were unstimulated or stimulated. So, which interacting proteins were found in these two conditions?
3. The data presented in Figure 3a would be more convincing if the authors provided a 3D render of IP3R1 distribution before and after stimulation. Or, at least, more images of the cells to show that they had taken appropriate z-sections. It is hard to imagine that all the IP3R1 within the cells migrates to the periphery upon stimulation.
4. The SIM images in Figure 3b are confusing. Why are they images of the interface of the cells and glass coverslips? They look like images through the middle of the cell to me. In addition, the brightness of the IP3R1 images for the Tespa1+/+ panels exceeds those for the Tespa1-/- and FFA panels, which gives a misleading impression of more translocation.
5. How were the data in Figure 3c obtained? What kind of background subtraction was performed? At first glance, it is surprising that there is strong significance between the Tespa1+/+ data and the FFAA data; they overlap considerably. The data for the CD3-stimulated cells be normalised against the mean fluorescence in unstimulated cells (to account for differences in IP3R1 expression). In which case, the FFAA data point would look very similar to the Tespa1+/+ data. In short, the data could be analysed to show that there is a Tespa1-independent change in fluorescence intensity. I would dispute the authors comment that 'These data strongly suggested that Tespa1 serves as an adaptor protein to recruit IP3R1 to TCR complexes after TCR activation.'
6. The different cytosolic calcium signals seen in response to ionomycin addition in calcium-free medium (Figure 5d) suggest that Tespa1 expression somehow affects the calcium content of intracellular stores. Why would this happen? This observation needs to be explained.
7. The IP3R2 antibody does not look as if it is working well. Where was it from? There are decent IP3R2 antibodies available.

There are numerous typographical errors and mistakes. Including the following:

1. Title. Change 'inositol 1, 4, 5-triphosphate receptors' to 'inositol 1,4,5-triphosphate receptors'.
2. Abstract. Line 32. Change 'Tespa1 elicit calcium...' to 'Tespa1 elicits calcium...'.
3. Abstract. Line 33. Change 'type 1 inositol-1, 4, 5- triphosphate...' to 'type 1 inositol 1,4,5-triphosphate...'.
4. Abstract. Line 39. Change 'Phenocoping...' to 'Phenocopying...'.
5. Line 139. Presumably 'family of ER proteins (Supplementary Fig. 1a).' should be 'family of ER proteins (Supplementary Fig. 2a).'
6. Sometimes the authors refer to cells as 'HEK', whereas other times they used different nomenclature (e.g. '293T cells'). Please be consistent.

Reviewer #2 (Remarks to the Author):

The paper by Liang and Lyu et al show that the protein Tespa1 is important for appropriate Ca²⁺ signaling in thymocytes, and that this is mediated through its interaction with inositol-trisphosphate receptor 1. They show that Tespa1 interacts with IP3R1 and that it apparently brings IP3R1 to the plasma membrane in the vicinity of TCR signaling. This interaction is necessary for T cell development in the thymus. This is a novel and interesting piece of work, thoroughly investigating the mechanism by which Tespa1 regulates IP3R1 and thus Ca²⁺ signaling in thymocytes and Jurkat. An earlier, very shallow study, had shown that Tespa1 binds to IP3R1 in T and B cells (ref 13), but I do not think that this detracts from the interest of this work.

Fig 1 shows convincingly that Tespa1 interacts with fragments of PLC β 1 –specifically the catalytic domain – and strongly suggests that this only occurs to a significant extent when the normal resting conformation of PLC β 1 is disrupted. This interaction is mediated through paired phenylalanine residues F187,188, as demonstrated by the fact that alanine mutations at these residues stop the interaction with IP3R1. It should be noted that the importance of these residues was predicted in Ref 10 – based on the similarity of Tespa1 to KRAS-induced actin-interacting protein (KRAP). This should be noted on line 140 (p7). Ref 10 is used inappropriately on line 91 (p5).

IP3R1 was redistributed to the plasma membrane region with WT Tespa1, but this did not occur when the FFAA mutant of residues 187, 188 was used (Fig 3a). The authors make the statement that IP3R1 and LAT colocalize (Line 159, p8). I don't think this is at all clear based on the red-green images presented. I don't see any yellow, which is what one would expect for colocalization. However, even if we did see some yellow in the image, there needs to be some image analysis to say that they are colocalized (in the same pixel of the image). Even so, this would not really be clear proof of interaction. To me it looks like IP3R1 is recruited to a region below the LAT signal. In this regard, Tespa1's association with IP3R1 (WT, not the FFAA mutant) is clear (there is lots of yellow in the image), but this is in the cytoplasm, not at the plasma membrane. (Fig 3a). This is in contrast to the data in the upper part of this fig where IP3R1 is clearly around the circumference of the cell, similar (though internal) to the LAT signal. The images do not show what the description says they do.

Images of thymocytes from WT mice, Tespa1 KO, or the KO with the FFAA transgene are shown in Fig3b, aiming to show that in the absence of Tespa1, less IP3R1 is recruited to the plasma membrane near the TCR (though again it is clearly below the membrane), and that the FFAA mutant transgene does not repair this defect. The differences are not very convincing from the images of individual cells presented, although the numbers in Fig 3c look good. What is MFI of IP3R1 in this context? Whole cell fluorescence or just around the plasma membrane? The data are interpreted to mean that Tespa1 is an adaptor bringing IP3R1 to the site of TCR signaling. In Line 180-183 on p9 they show that the pY-353 signal of IP3R1 is increased significantly after TCR

stimulation, but it is not fair to state that it shows "dramatic colocalization with LAT" without any quantitation, because it was somewhat colocalized with LAT even before stimulation (and again, the data are strange because LAT is all cytoplasmic in these images (Fig 4a). The FFAA doesn't show much pY-353 signal before or after stimulation. The same goes for what they say about pY-353 signal in thymocytes. It is not visibly colocalized (no yellow at all). To my eye it is predominantly below the TCR. They really need to be measuring colocalization, for example with Pearson correlation coefficient analysis.

The Ca²⁺ flux data (Fig 5) are convincing in showing that Tespa1 is important in giving increased ER Ca²⁺ release and NFAT and AP1 activity, but that NFκB activity is not affected. The FFAA mutant doesn't do this. Ca²⁺ flux is also shown to occur in the same region as the TCR using TIRFM (Fig 6). I don't think it is fair to call these "Ca²⁺ microdomains" as the Ca²⁺ signal is much more diffuse than the TCR signal, and there is no attempt to show if the signals are close in space.

These data suggest that the ER Ca²⁺ release occurs close to the TCR, presumably near the LAT signalosome. Why? Is it to also lead to SOCE in the region? Presumably this would not involve ORAI but some other Ca²⁺ transporters. Some discussion of this point would be worthwhile.

The weakest part of the paper is the description and analysis of the imaging data. This needs to be addressed.

Minor points (with page and line numbers):

2. 32. elicits, not elicit.
2. 39. phenocoping should be phenocopying
4. 91. Ref 10 is used incorrectly! This ref correctly predicted that Tespa1 would interact with IP3R1 based on homology to KRAP. Ref 13 is more appropriate for human tespa1 KRAP data. In Discussion line 297 Ref 10 is used correctly.
6. 132. A bad sentence. I think they probably mean "...by which Tespa1 regulates TCR signaling...".
7. 156. trans-localization should be translocation.
15. 313. redistributes
15. 318-319. "We also showed that the recruitment of IP3R1 to the TCR complex is essential for the Y353 phosphorylation of IP3R1 by Fyn". This is not accurate, as this paper does not look at Fyn activity at all.

Point-by-point Responses:

Reviewer #1 (Remarks to the Author):

This is a potentially interesting and novel study. The authors use a range of approaches to demonstrate the role of Tespa1 in regulating TCR-mediated calcium signals. However, it was not an easy read and not fully convincing; the figures were not well described in the text, and the figure legends did not describe the experimental data at all well. I had to stare at the figures for a while to work out what was going on and correlate it with the authors' statements. I am not sure that many people will give the manuscript so much time.

For example, Panel d in figure 1 is entirely explained by the sentence (line 126) 'This finding supported our previous finding that Tespa1 binds to the LAT signalosome upon TCR activation (Fig. 1d).' The data is not solely a previous finding! It's in the current manuscript, and it needs to be described!

Some of the figures are not labelled correctly. For example, Panels a and b in Figure 2 have the same labelling, but the IPs and blots were performed with different antibodies (to show a co-IP). Please check the figures carefully. It's very confusing for the reader otherwise.

The authors may have discovered a novel role for Tespa1 in linking signalling proteins downstream of TCR stimulation, but I have really struggled with the manuscript in terms of data quality, explanation of how data were obtained, explanation of the figures and control experiments.

Major comments:

1. I don't find the immunoblots in Figure 1 to be fully convincing of the points the authors make. On line 128, the authors state 'In sum, these results demonstrated that the recruitment of Tespa1 to the LAT signalosome upon TCR activation is mediated by interaction with the exposed catalytic domain of PLC- γ 1.' However, this is not directly shown by the data. In fact, there is little (or none) interaction of full-length PLC- γ 1 with Tespa1 according to the data shown. Only deletion mutations, or point mutations, increase binding. Whilst this is consistent with the authors' suggestions, it is not proof. Does co-IP of over-expressed proteins in HEK cells does replicate TCR organization in thymocytes? I'm not convinced it does. Panel d in figure 1 indicates a receptor-activated interaction that is again consistent with the authors' suggestion. However, I think the authors should tone down their conclusion from these data.

Response:

We thank the reviewer for pointing this out. We agree with the reviewer's comments and would like to take the suggestion and rephrase the conclusion for Fig.1. We would like to change our conclusion to **'In summary, these results suggest that the recruitment of Tespa1 to the LAT signalosome upon TCR activation might be mediated by the interaction between Tespa1 and PLC- γ .'** (line 129 on p6). Regarding to the question of cell system, although the overexpression and Co-IP experiments in HEK293 cells could not fully represent the Tespa1-PLC- γ 1 interactions in T cells, this is a generalized research method to comprehend protein-protein

interaction details, and is based on our observations of this interaction in T cells (Fig. 1c, and the Fig.9b of the article: Di Wang, et al, Nat. Immunol. 2012 May 6; 13(6): 560-568¹).

2. Was there are difference in the Tespa1 interactome in unstimulated versus stimulated cells? The main text does not say so, but the legend to Supplementary Figure 2 states that cells were unstimulated or stimulated. So, which interacting proteins were found in these two conditions?

Response:

We are sorry for the confusion. The Tespa1 interactome analysis showed in Supplementary Figure 2 was carried out in anti-CD3 stimulated Jurkat cells. We have incorporated this information in the revised figure legend. We have performed the same analysis in unstimulated cells, but fail to identify any substantial binding partners. It is reasonable for us because we later found in our verification experiments that the binding of Tespa1 to its interacting partners largely depends on the activation of TCR.

3. The data presented in Figure 3a would be more convincing if the authors provided a 3D render of IP3R1 distribution before and after stimulation. Or, at least, more images of the cells to show that they had taken appropriate z-sections. It is hard to imagine that all the IP3R1 within the cells migrates to the periphery upon stimulation.

Response:

The data showed in Fig 3a were acquired by confocal microscopy. Because of low resolution of the images it is not possible to show the precise locations of the stained molecules. To overcome this problem, we have performed Structured Illumination Microscopy (SIM) to reconfirm the recruitment of IP3R1 to TCR complex. The 3D-display of IP3R1 (Fig 3b) and IP3R1 pY353 (S-figure 4b-2) distribution is now included to give the whole-cell distributions of the molecules before and after TCR activation. Although there are no significant changes of overall IP3R1 distribution in the whole cells, the recruitment of IP3R1 to the interface and its phosphorylation on the Y353 after stimulation occurred obviously in S-figure 4b-2 and movie rs2.

S-figure 4b-2. 3D display of phosphorylated IP3R. WT thymocytes were stimulated with (10min) or without (0min) anti-CD3 coated coverslip. Images were acquired at 4um Z stack from the focal plane and reconstituted in 3D display

4. The SIM images in Figure 3b are confusing. Why are they images of the interface of the cells and glass coverslips? They look like images through the middle of the cell to me.

Response:

These pictures were taken from the interface between the cells and the anti-CD3 antibody-coated coverslips. To identify the focal plane, we used the nonspecific binding signals between the second antibody and the coverslips. Images of IP3R1 clusters were then collected on the same focal plane. The Diagram was shown in **Supplementary Fig. 8** and supplemental **movie rs1**. Detail **methods** are now included in the revised manuscript **(line 448 on p21)**.

In addition, the brightness of the IP3R1 images for the Tespa1+/+ panels exceeds those for the Tespa1-/- and FFA panels, which gives a misleading impression of more translocation.

Response:

A fixed LUT(Look up table) value was used to display all the images in the same experiment, thus the intensity of signal reflects the translocation of IP3R1 to the focal plane, as summarized in Fig 3d.

5. How were the data in Figure 3c obtained? What kind of background subtraction was performed? At first glance, it is surprising that there is strong significance between the Tespa1+/+ data and the FFAA data; they overlap considerably.

Response:

The images were obtained in the interface of the cells and coverslips. Detail **methods** are now included in the revised manuscript **(line 448 on p21)**. Images processing procedure is: use Metamorph software to open the tiff file, call 'integrated morphometry analysis', 'region measurements', 'threshold image' programs. The 'integrated morphometry analysis' is used for measuring the intensity of ip3r/pY353, the 'region measurements' is used for measuring the area of interest, the 'threshold image' is used for wiping off background, and our threshold value is 1100. ROI means the region of interest. We draw ROI according to the outline of the interface. The MFI is calculated as: $MFI = (\text{total intensity})/ROI$.

In this particular experiment showed in Fig 3c, the difference between the FFAA group and WT was smaller than the difference between the KO, but it is still statistically significant ($P < 0.01$). To make sure if the difference is typical or not, we repeated the experiment and got a similar result. The difference between the wild-type and other two groups are consistent, while the difference between the KO and FFAA groups is not statistically different although the values vary between experiments. We now replaced the **figure 3c** with our new results (**Figure 3d**).

The data for the CD3-stimulated cells be normalized against the mean fluorescence in unstimulated cells (to account for differences in IP3R1 expression). In which case, the FFAA data point would look very similar to the *Tespa1*^{+/+} data. In short, the data could be analyzed to show that there is a *Tespa1*-independent change in fluorescence intensity. I would dispute the authors comment that ‘These data strongly suggested that *Tespa1* serves as an adaptor protein to recruit IP3R1 to TCR complexes after TCR activation.’

Response:

We detected a comparable protein expression level of IP3R1 in WT, KO and FFAA thymocytes (**S-Figure 3b-3**). Moreover, our confocal data in Fig 3a also showed a similar IP3R1 fluorescence intensity in both unstimulated J-*Tespa1* and J-FFAA cells. So we believe that *Tespa1* has limited effect on the expression of IP3R1.

The lower level of IP3R1 fluorescence intensity on the interface, which reflected the defective redistribution of IP3R1 to the plasma membrane region in resting FFAA cells, might be due to its defective binding capability with *Tespa1*. Furthermore the physiological role of this basal relocation of IP3R1 in resting cells is still elusive for us at this stage. However, this *Tespa1*-dependent relocation of IP3R1 appears more significant after TCR activation, which is believed to be important for the optimal downstream TCR activation.

S-Figure 3b-3. Immunoblots (IB) of IP3R1 in sorted DP thymocytes from WT (*Tespa1*^{+/+}), KO (*Tespa1*^{-/-}), and FFAA (*Tespa1*-FFAA) mice. 1 and 2 indicate two different mice.

6. The different cytosolic calcium signals seen in response to ionomycin addition in calcium-free medium (Figure 5d) suggest that Tespa1 expression somehow affects the calcium content of intracellular stores. Why would this happen? This observation needs to be explained.

Response:

It is true that decreased calcium flux were observed not only in TCR stimulated cells but also upon PMA+ionomycin, which suggested a possible role of Tespa1 in the intracellular calcium store (Fig. 8b,c of the article: *Di Wang et al, Nat. Immunol. 2012 May 6; 13(6): 560-568¹.*). Since it has been reported that altered IP3R function might result in disrupted cellular Ca²⁺ homeostasis and decreased intracellular Ca²⁺ stores, reduced basal IP3R1 activation in Tespa1-deficient cells might be responsible for this phenomenon (*Kopil C. M. et al, J. Biol. Chem. 2011 Oct 14; 286, 35998–36010²; Tomohiro NAKAYAMA et al, Biochem. J. 2004 Jan 15; 377, 299–307³*). However, we could not rule out the possibility that Tespa1 might have other unidentified functions in regulating the ER calcium store or be involved in PMA-ionomycin-induced calcium flux through an unknown mechanism. Now this discussion has been incorporated in the manuscript (line 226 on p11).

7. The IP3R2 antibody does not look as if it is working well. Where was it from? There are decent IP3R2 antibodies available.

Response:

The anti-IP3 Receptor Type II Antibody used in our experiment was purchased from Merck Millipore, Catalogue Number: AB3000. We have checked the antibody using protein lysates from different cells including RBL-2H3 and BMMCs (Bone-marrow derived mast cells) where IP3R2 is known to express at high level. It is obvious that the expression of IP3R2 is very low in both Jurkat T cells and thymocytes (S-Figure 2c-2). Thus, the weak signal of IP3R2 in the manuscript reflects the low expression of the IP3R2.

S-Figure 2c-2. Immunoblots (IB) of IP3R2 in thymocytes, RBL-2H3 cell, BMMCs, Jurkat cells, NIH3T3 cells and RAW264.7 cells.

There are numerous typographical errors and mistakes. Including the following:

1. Title. Change ‘inositol 1, 4, 5-triphosphate receptors’ to ‘inositol 1,4,5-trisphosphate receptors’.
2. Abstract. Line 32. Change ‘Tespa1 elicit calcium...’ to ‘Tespa1 elicits calcium...’.
3. Abstract. Line 33. Change ‘type 1 inositol-1, 4, 5- triphosphate...’ to ‘type 1 inositol 1,4,5- trisphosphate...’.
4. Abstract. Line 39. Change ‘Phenocoping...’ to ‘Phenocopying...’.
5. Line 139. Presumably ‘family of ER proteins (Supplementary Fig. 1a).’ should be ‘family of ER proteins (Supplementary Fig. 2a).’
6. Sometimes the authors refer to cells as ‘HEK’, whereas other times they used different nomenclature (e.g. ‘293T cells’). Please be consistent.

Response:

We thank the reviewer for pointing out the errors in our writing, we have corrected them in our revised manuscript. In order to further improve the language of the manuscript, we have used the Springer Nature Author Services to edit the revised manuscript.

Reviewer #2 (Remarks to the Author):

The paper by Liang and Lyu et al show that the protein Tespa1 is important for appropriate Ca²⁺ signaling in thymocytes, and that this is mediated through its interaction with inositol-tris-phosphate receptor 1. They show that Tespa1 interacts with IP3R1 and that it apparently brings IP3R1 to the plasma membrane in the vicinity of TCR signaling. This interaction is necessary for T cell development in the thymus. This is a novel and interesting piece of work, thoroughly investigating the mechanism by which Tespa1 regulates IP3R1 and thus Ca²⁺ signaling in thymocytes and Jurkat. An earlier, very shallow study, had shown that Tespa1 binds to IP3R1 in T and B cells (ref 13), but I do not think that this detracts from the interest of this work.

1. Fig 1 shows convincingly that Tespa1 interacts with fragments of PLCg1 –specifically the catalytic domain – and strongly suggests that this only occurs to a significant extent when the normal resting conformation of PLCg1 is disrupted. This interaction is mediated through paired phenylalanine residues F187, 188, as demonstrated by the fact that alanine mutations at these residues stop the interaction with IP3R1. It should be noted that the importance of these residues was predicted in Ref 10 – based on the similarity of Tespa1 to KRAS-induced actin-interacting protein (KRAP). This should be noted on line 140 (p7). Ref 10 is used inappropriately on line 91 (p5).

Response:

We thank the reviewer for pointing out this issue. We have reorganized our statement and the sequence of the reference based on your advice.

2. IP3R1 was redistributed to the plasma membrane region with WT Tespa1, but this did not occur when the FFAA mutant of residues 187, 188 was used (Fig 3a). The authors make the statement that IP3R1 and LAT colocalize (Line 159, p8). I don't think this is at all clear based on the red-green images presented. I don't see any yellow, which is what one would expect for colocalization. However, even if we did see some yellow in the image, there needs to be some image analysis to say that they are colocalized (in the same pixel of the image). Even so, this would not really be clear proof of interaction. To me it looks like IP3R1 is recruited to a region below the LAT signal. In this regard, Tespa1's association with IP3R1 (WT, not the FFAA mutant) is clear (there is lots of yellow in the image), but this is in the cytoplasm, not at the plasma membrane. (Fig 3a). This is in contrast to the data in the upper part of this fig where IP3R1 is clearly around the circumference of the cell, similar (though internal) to the LAT signal. The images do not show what the description says they do.

Response:

We totally agree with the reviewer's comments. We are sorry for the confusions. The exact point we want to make is: upon TCR activation, there is re-localization of IP3R1 close to the plasma membrane and LAT in a Tespa1-dependent manner. This is further clarified in the SIM experiment with higher resolution. Thus it is not suitable to use colocalization in our conclusion. We have changed our statement of Fig. 3a to 'However, after TCR stimulation for 5 min, IP3R1 staining was redistributed and localized in proximity to LAT in J-Tespa1 cells but not in J-FFAA cells' (line160 on p8).

3. Images of thymocytes from WT mice, Tespa1 KO, or the KO with the FFAA transgene are shown in Fig3b, aiming to show that in the absence of Tespa1, less IP3R1 is recruited to the plasma membrane near the TCR (though again it is clearly below the membrane), and that the FFAA mutant transgene does not repair this defect. The differences are not very convincing from the images of individual cells presented, although the numbers in Fig 3c look good. What is MFI of IP3R1 in this context? Whole cell fluorescence or just around the plasma membrane? The data are interpreted to mean that Tespa1 is an adaptor bringing IP3R1 to the site of TCR signaling.

Response:

Fig 3c summarized the MFI of IP3R1 around the plasma membrane at the interface between the cells and anti-CD3 coated coverslips. To help better understand the data, detail methods are included in the methods section of the paper (line 448 on p21). Images processing procedure is: use Metamorph software to open the tiff file, call 'integrated morphometry analysis', 'region measurements', 'threshold image' programs. The 'integrated morphometry analysis' is used for measuring the intensity of ip3r/pY353, the 'region measurements' is used for measuring the area of interest, the 'threshold image' is used for wiping off background, and our threshold

value is 1100. ROI means the region of interest. We draw ROI according to the outline of the interface. The MFI is calculated as: $MFI = (\text{total intensity}) / \text{ROI}$.

We've repeated the experiment for more than 3 times and collected data from hundreds of cells, the difference between WT and KO/FFAA is consistent and statistically significant. We agree that this data is not sufficient to draw the conclusion that 'Tesp1 is an adaptor bringing IP3R1 to the site of TCR signaling', we have changed it to 'These data suggested the potential role of Tespa1 in mediating the relocalization of IP3R1 to TCR-proximal region after TCR activation.' in the revised manuscript (line 175-177 on p8).

4. In Line 180-183 on p9 they show that the pY-353 signal of IP3R1 is increased significantly after TCR stimulation, but it is not fair to state that it shows "dramatic colocalization with LAT" without any quantitation, because it was somewhat colocalized with LAT even before stimulation (and again, the data are strange because LAT is all cytoplasmic in these images (Fig 4a). The FFAA doesn't show much pY-353 signal before or after stimulation. The same goes for what they say about pY-353 signal in thymocytes. It is not visibly colocalized (no yellow at all). To my eye it is predominantly below the TCR. They really need to be measuring colocalization, for example with Pearson correlation coefficient analysis.

Response:

As responded above, we would like to change our statement from 'colocalization' to 'relocalized close to' (line 183-186 on p9). The poor resolution of confocal image makes it hard to tell the exact location of LAT in Fig 4a. This is one of the reasons why we further performed SIM experiment in Fig 4b. We looked back to our original data, the strong pY353 signal and relocalization to LAT in unstimulated cells is not typical. It is probably due to the higher basal activation of these cells. We have now replaced that panel with a more representative image.

5. The Ca^{2+} flux data (Fig 5) are convincing in showing that Tespa1 is important in giving increased ER Ca^{2+} release and NFAT and AP1 activity, but that NFkB activity is not affected. The FFAA mutant doesn't do this. Ca^{2+} flux is also shown to occur in the same region as the TCR using TIRFM (Fig 6). I don't think it is fair to call these " Ca^{2+} microdomains" as the Ca^{2+} signal is much more diffuse than the TCR signal, and there is no attempt to show if the signals are close in space. These data suggest that the ER Ca^{2+} release occurs close to the TCR, presumably near the LAT signalosome. Why? Is it to also lead to SOCE in the region? Presumably this would not involve ORAI but some other Ca^{2+} transporters. Some discussion of this point would be worthwhile.

Response:

We thank the reviewer for the suggestion. The " Ca^{2+} microdomains" is used by Xiaoshan Shi in describing the regional Ca^{2+} (Xiaoshan Shi et al, Nature 2013, 493(7430): 111-115.⁴). It is true that the condition used in our experiment is not enough to define those microdomains. We agree to avoid using this term " Ca^{2+} microdomains" in our manuscript. The point we want to make

based on the obtained results is that the recruitment of IP3R1 close to the TCR by Tespa1 along with TCR stimulation might lead to faster and stronger activation of IP3R1 and Ca²⁺ release close to the TCR as discussed in the manuscript (line 335-340 on p16). We concluded that the ER Ca²⁺ release occurs close to the TCR, since the calcium signal overlap with the TCR signal in WT cells. While in the KO and FFAA cells, the calcium signal was not only much weaker in the intensity but also less associated with TCR clusters.

All the Ca²⁺ imaging experiments were performed in Ca²⁺-free medium to exclude the influence of store-operated calcium channels (SOCs) or other Ca²⁺ channels on the plasma membrane; thus, the Ca²⁺ signal captured in this experiment was mediated only by IP3Rs. In addition, store-operated Ca²⁺ influx did not differ among WT, KO, and Tespa1-FFAA thymocytes (Fig 5c,d, line 221-223 on p10). This is now discussed in the manuscript (line 250-254 on p12).

Minor points (with page and line numbers):

2. 32. elicits, not elicit.

2. 39. phenocoping should be phenocopying

4. 91. Ref 10 is used incorrectly! This ref correctly predicted that Tespa1 would interact with IP3R1 based on homology to KRAP. Ref 13 is more appropriate for human tespa1 KRAP data. In Discussion line 297 Ref 10 is used correctly.

6. 132. A bad sentence. I think they probably mean "...by which Tespa1 regulates TCR signaling..."

7. 156. trans-localization should be translocation.

15. 313. redistributes

15. 318-319. "We also showed that the recruitment of IP3R1 to the TCR complex is essential for the Y353 phosphorylation of IP3R1 by Fyn". This is not accurate, as this paper does not look at Fyn activity at all.

Response:

We thank the reviewer for pointing out the errors in our writing, we have corrected them in our revised manuscript. In order to further improve the language of the manuscript, we have used the Springer Nature Author Services to edit the revised manuscript.

Reference:

1. Wang D, Zheng M, Lei L, Ji J, Yao Y, Qiu Y, *et al.* Tespa1 is involved in late thymocyte development through the regulation of TCR-mediated signaling. *Nature immunology* 2012, **13**(6): 560-568.
2. Kopil CM, Vais H, Cheung KH, Siebert AP, Mak DO, Foskett JK, *et al.* Calpain-cleaved type 1 inositol 1,4,5-trisphosphate receptor (InsP(3)R1) has InsP(3)-independent gating and disrupts intracellular Ca(2+) homeostasis. *The Journal of biological chemistry* 2011, **286**(41): 35998-36010.
3. Assefa Z, Bultynck G, Szlufcik K, Nadif Kasri N, Vermassen E, Goris J, *et al.* Caspase-3-induced truncation of type 1 inositol trisphosphate receptor accelerates apoptotic cell death and induces inositol trisphosphate-independent calcium release during apoptosis. *The Journal of biological chemistry* 2004, **279**(41): 43227-43236.
4. Shi X, Bi Y, Yang W, Guo X, Jiang Y, Wan C, *et al.* Ca²⁺ regulates T-cell receptor activation by modulating the charge property of lipids. *Nature* 2013, **493**(7430): 111-115.

Reviewers' comments:

Reviewer #1 (Remarks to the Author):

The authors have responded with consideration to my queries. I still have reservations about the quality of the data, how it is explained and the interpretation. I think the authors have interesting and novel data, but I am just not fully convinced. We could sit and chat for ages about the interpretation of many of the figures. Moreover, the statistical significances shown are all based on multiple t-tests, which is incorrect. The authors should consider using ANOVA at the very least. ANOVA is more conservative, and I imagine the significance will not be the same.

1. Incredibly, the title is still wrong! 'inositol 1,4,5-triphosphate receptors' should be 'inositol 1,4,5-trisphosphate receptors'.

2. Line 34. 'inositol-1,4,5-triphosphate receptor' should be 'inositol 1,4,5-trisphosphate receptor'.

3. Line 140. I don't understand why the sentence 'Interestingly, the importance of these residues was predicted by Nicholas R J Gascoigne based on the similarity of Tespa1 to KRAS-induced actin-interacting protein (KRAP)' is inserted into the Results. I think some text concerning F187 and F188 might be missing at this point.

4. Line 158. I'm afraid that I am still not convinced that IP3R1 translocates to LAT complexes. The 0 min images in Figure 3a look like a different cellular plane to me. I honestly can't decide if I am convinced by the SIM images. The authors claim to be looking at the cell:coverslip interface, but frankly it's hard to tell. Why didn't the authors use TIRF, as then there would be no confusion about focal depth

5. Figure 3d suggests that there was significant translocation of IP3R1 in the absence of Tespa1 and with the FAA mutant. This should be discussed. Given the spread of data shown in Figure 3d, I find it hard to understand how $P < 0.001$ was obtained. The means are different, but the data ranges are virtually the same. Using multiple t-tests for comparison of the different populations is not valid.

Reviewer #2 (Remarks to the Author):

This paper is interesting and worthwhile. However, the inconsistencies and mistakes in the writing are extremely frustrating.

This statement is part of the description of Figure 3:

"Although there were no significant changes in the overall IP3R1 distribution in whole cells (Fig.3b) after TCR activation, the 3D display showed that Tespa1-dependent recruitment of IP3R1 appeared to occur at the interface after stimulation (Fig. 3c, d)."

Firstly, it is unclear – it refers to a 3D display that I don't see and to Fig 3d that isn't there. Are the images in 3b (from SIM) a cross-section through the cell or are they images of the interface between the T cell and the Ab-coated cover slip? According to the rebuttal they are of the interface. In that case, why is all the TCR in the outer ring. And indeed, why is the nucleus so visible? I don't understand – either the description is wrong or it is incomplete.

Regarding Fig 4. The rebuttal states "As responded above, we would like to change our statement from 'colocalization' to 'relocalized close to' (line 183-186 on p9). The poor resolution of confocal image makes it hard to tell the exact location of LAT in Fig 4a. This is one of the reasons why we further performed SIM experiment in Fig 4b." I am not convinced that it is appropriate to say "close to" here. The red and green dots are fairly close in the images, but I do not think that this necessarily translates to them being close at the molecular scale. How about saying "relocalized closer to"?

There are 3 movies. Only one is mentioned in the text, as Supplemental movie 1. As far as I can tell, it is actually the 3rd movie. What are the others?

The new version still makes a hash of the use of the references that are currently numbered 10 and 13. As I pointed out in the initial review, the statement on Line 90, p4. "Similarly, it was recently reported that human Tespa1 protein interacts with IP3R1 and regulates Ca²⁺ signaling (ref 10)" is incorrect. Ref 10 is Gascoigne NR, Fu G. Tespa1: another gatekeeper for positive selection. *Nature immunology* 2012, 13(6): 530-532. The correct reference for the statement is currently #13: Matsuzaki H, Fujimoto T, Ota T, Ogawa M, Tsunoda T, Doi K, et al. Tespa1 is a novel inositol 1,4,5-trisphosphate receptor binding protein in T and B lymphocytes. *FEBS open bio* 2012, 2: 255-259. Reference 10 correctly predicted that because Tespa1 is closely related to KRAP, that it would interact with IP3R, and that it would do so through its F187, F188 residues.

I therefore propose the following edit to this section of the Introduction:
Starting on line 90:

".....ERK-AP-1 and Ca²⁺-NFAT pathways. The similarity of Tespa1 to Ki-Ras-induced actin-interacting protein (KRAP) led to the prediction that Tespa1 would interact with IP3R (current ref 10), and it was recently reported that human Tespa1 protein interacts with IP3R1 and regulates Ca²⁺ signaling (current ref 13)."

The section starting on line 140, p7 which credits the predictions of ref #10 actually references the wrong paper: #13 instead of #10. I therefore propose the following edit:

"...and 3, a family of ER proteins (Supplementary Fig. 2a). Interestingly, this interaction, and the importance of the residues F187 and F188 in mediating the interaction, was predicted based on the sequence similarity of Tespa1 to KRAS-induced actin-interacting protein (KRAP) (current Ref 10). We next verified the interaction between Tespa1 and IP3Rs, and the role of F187 and F188 sites in mediating this interaction."

Point-by-point Responses:

Reviewer #1 (Remarks to the Author):

The authors have responded with consideration to my queries. I still have reservations about the quality of the data, how it is explained and the interpretation. I think the authors have interesting and novel data, but I am just not fully convinced. We could sit and chat for ages about the interpretation of many of the figures. Moreover, the statistical significances shown are all based on multiple t-tests, which is incorrect. The authors should consider using ANOVA at the very least. ANOVA is more conservative, and I imagine the significance will not be the same.

Response:

We thank the reviewer for the comments and suggestions. We've presented and discussed our story with many scientists in the field under different circumstances including in the AAI meeting. We hope that we will have the opportunity to discuss this story, as well as our other researches, with the reviewer in the future. We've changed the statistical analysis using ANOVA as suggested by the reviewer and the differences between our experimental groups are still significant.

1. Incredibly, the title is still wrong! 'inositol 1,4,5-triphosphate receptors' should be 'inositol 1,4,5-trisphosphate receptors'.
2. Line 34. 'inositol-1,4,5-triphosphate receptor' should be 'inositol 1,4,5-trisphosphate receptor'.

Response:

We are very sorry for the mistakes. This is probably due to the mixing up of different versions during the revision of our manuscript. We have corrected it.

3. Line 140. I don't understand why the sentence 'Interestingly, the importance of these residues was predicted by Nicholas R J Gascoigne based on the similarity of Tespa1 to KRAS-induced actin-interacting protein (KRAP)' is inserted into the Results. I think some text concerning F187 and F188 might be missing at this point.

Response:

We are sorry for the confusion. We have now changed the sentence to 'Interestingly, this interaction, and the importance of the residues F187 and F188 in mediating the interaction, was predicted based on the sequence similarity of Tespa1 to Ki-Ras-induced actin-interacting protein (KRAP)' (line 142-145) as suggested by the other reviewer.

4. Line 158. I'm afraid that I am still not convinced that IP3R1 translocates to LAT complexes. The 0

min images in Figure 3a look like a different cellular plane to me. I honestly can't decide if I am convinced by the SIM images. The authors claim to be looking at the cell:coverslip interface, but frankly it's hard to tell. Why didn't the authors use TIRF, as then there would be no confusion about focal depth.

Response:

We have performed TIRF experiments as requested by the reviewer and obtained very similar results. It clearly showed the recruitment of IP3R to the interface between the T cell and the anti-CD3 coated glass coverslips, which is impaired in the *Tespa1*^{-/-} and *Tespa1*-FFAA T cells. We have now included this data into the revised manuscripts as supplementary results (supplementary Fig. 4 and 5) (line178-182 and 200-203). Interestingly, a recent paper in the newest issue of *Nat Immunology* ("Recruitment of calcineurin to the TCR positively regulates T cell activation" 2016 Dec 12.) used similar method to stimulate T cells with anti-CD3 antibodies coated on glass coverslips. Since all three imaging experiments provided us similar results, we believe our SIM data are convincing.

5. Figure 3d suggests that there was significant translocation of IP3R1 in the absence of *Tespa1* and with the FAA mutant. This should be discussed. Given the spread of data shown in Figure 3d, I find it hard to understand how $P < 0.001$ was obtained. The means are different, but the data ranges are virtually the same. Using multiple t-tests for comparison of the different populations is not valid.

Response:

We thank the reviewer for the suggestion. This is now discussed in the manuscript (line182-186). We have used ANOVA and Mann-Whitney U test for the statistical analysis and got same conclusions, there are significant difference of IP3R1 and pY353 displays between *Tespa1*^{-/-}/FFAA and WT.

Reviewer #2 (Remarks to the Author):

This paper is interesting and worthwhile. However, the inconsistencies and mistakes in the writing are extremely frustrating.

This statement is part of the description of Figure 3:

"Although there were no significant changes in the overall IP3R1 distribution in whole cells (Fig.3b) after TCR activation, the 3D display showed that *Tespa1*-dependent recruitment of IP3R1 appeared to occur at the interface after stimulation (Fig. 3c, d)."

Firstly, it is unclear – it refers to a 3D display that I don't see and to Fig 3d that isn't there. Are the images in 3b (from SIM) a cross-section through the cell or are they images of the interface between the T cell and the Ab-coated cover slip? According to the rebuttal they are of the interface. In that case, why is all the TCR in the outer ring. And indeed, why is the nucleus so visible? I don't

understand – either the description is wrong or it is incomplete.

Response:

We are sorry for the confusion. The images in 3b (from SIM) are 3D display images, and the images in Fig 3c were obtained from the interface between the cells and the cover slips. The images in Fig 3b are used to construct the different overall distribution of the IP3R1 in resting and activated T cells. The red fluorescence in the Fig3b is not TCR but IP3R1. The left two images are the cross-sections through the cell, the IP3R1 is in the outer ring (the cytoplasm) and the nucleus is visible. The right two images are the Z-axis of the cells, in which we can see the overall IP3R1 distribution in the whole cells.

We have clarified this statement in the manuscript: ‘Although the 3D display (Fig. 3b) showed that there were no significant changes in the overall IP3R1 distribution in the whole cells after TCR activation, Tespa1-dependent recruitment of IP3R1 appeared to occur at the interface after stimulation (Fig. 3c, d).’ (line174-178).

We notice that the misunderstanding is most likely due to our incomplete description in the figure legend. We have now rewritten the figure legend for figure 3bL: ‘A 4 um Z-stack 3D display was taken from contacting coverslip by Structured illumination microscopy (SIM). Left, a view from side of contacting coverslip. Right, a view upright to Z axis.’ (line718-721).

Regarding Fig 4. The rebuttal states “As responded above, we would like to change our statement from ‘colocalization’ to ‘relocalized close to’ (line 183-186 on p9). The poor resolution of confocal image makes it hard to tell the exact location of LAT in Fig 4a. This is one of the reasons why we further performed SIM experiment in Fig 4b.” I am not convinced that it is appropriate to say “close to” here. The red and green dots are fairly close in the images, but I do not think that this necessarily translates to them being close at the molecular scale. How about saying “relocalized closer to”?

Response:

We thank the reviewer for the suggestion, and agree with the reviewer to make this change (line196).

There are 3 movies. Only one is mentioned in the text, as Supplemental movie 1. As far as I can tell, it is actually the 3rd movie. What are the others?

Response:

The other two movies were used to explain the questions of the other reviewer, which will not be included in the manuscript.

The new version still makes a hash of the use of the references that are currently numbered 10 and 13.

As I pointed out in the initial review, the statement on Line 90, p4. “Similarly, it was recently reported that human Tespa1 protein interacts with IP3R1 and regulates Ca²⁺ signaling (ref 10)” is incorrect. Ref 10 is Gascoigne NR, Fu G. Tespa1: another gatekeeper for positive selection. Nature immunology 2012, 13(6): 530-532. The correct reference for the statement is currently #13: Matsuzaki H, Fujimoto T, Ota T, Ogawa M, Tsunoda T, Doi K, et al. Tespa1 is a novel inositol 1,4,5-trisphosphate receptor binding protein in T and B lymphocytes. FEBS open bio 2012, 2: 255-259. Reference 10 correctly predicted that because Tespa1 is closely related to KRAP, that it would interact with IP3R, and that it would do so through its F187, F188 residues.

I therefore propose the following edit to this section of the Introduction:
Starting on line 90:

“.....ERK-AP-1 and Ca²⁺-NFAT pathways. The similarity of Tespa1 to Ki-Ras-induced actin-interacting protein (KRAP) led to the prediction that Tespa1 would interact with IP3R (current ref 10), and it was recently reported that human Tespa1 protein interacts with IP3R1 and regulates Ca²⁺ signaling (current ref 13).”

The section starting on line 140, p7 which credits the predictions of ref #10 actually references the wrong paper: #13 instead of #10. I therefore propose the following edit:

“...and 3, a family of ER proteins (Supplementary Fig. 2a). Interestingly, this interaction, and the importance of the residues F187 and F188 in mediating the interaction, was predicted based on the sequence similarity of Tespa1 to KRAS-induced actin-interacting protein (KRAP) (current Ref 10). We next verified the interaction between Tespa1 and IP3Rs, and the role of F187 and F188 sites in mediating this interaction.”

Response:

We are very sorry for the mistake and extremely thankful to the reviewer for the help on these editing. They are now included in the manuscript (line90-94, 1420-145). We have corrected the reference order.

REVIEWERS' COMMENTS:

Reviewer #2 (Remarks to the Author):

The authors have responded appropriately to the second reviews.

Reviewers' comments and point-by-point responses:

Reviewer #2 (Remarks to the Author):

The authors have responded appropriately to the second reviews.

Response:

We thank the reviewer for his/her help and we are glad that our response satisfied him/her.